# A Distributional Analogue to the Successor Representation

## Abstract

This paper contributes a new approach for distributional reinforcement learning which allows for a clean separation of transition structure and reward in the learning process. Analogous to how the successor representation (SR) describes the expected consequences from behaving according to a given policy, our distributional successor measure (SM) describes the distributional consequences of this behaviour. We model the distributional SM as a distribution over distributions and provide theory connecting it with distributional and model-based reinforcement learning. Extending $\gamma$-models (Janner et al., 2020), we propose an algorithm that learns the distributional successor measure from samples by minimizing a two-level maximum mean discrepancy. Key to our method are a number of algorithmic techniques that are independently valuable in the context of learning generative models of state. As an illustration of the practical usefulness of the distributional successor measure, we show that it enables zero-shot risk-sensitive policy evaluation in a way that was not previously possible.

## 1 Introduction

Distributional reinforcement learning (Morimura et al., 2010; Bellemare et al., 2017a; 2023) is an approach to reinforcement learning (RL) that focuses on learning the entire probability distribution of an agent's return, not just its expected value. Distributional RL has been shown to improve deep RL agent performance (Yang et al., 2019; Nguyen-Tang et al., 2021), and provides a flexible approach to risk-aware decision-making (Dabney et al., 2018a; Fawzi et al., 2022; Zhang & Weng, 2021). A notable drawback of existing approaches to distributional RL is that rewards must be available at training time in order to predict the return distribution. In particular, if we wish to evaluate a trained policy on a new reward function, to understand the behaviour of the policy for various performance criteria, for example, these predictions of the return distributions must be trained from scratch. This paper contributes a methodology that overcomes this drawback, and allows for *zero-shot evaluation* of novel reward functions without requiring further learning.

In the case of predicting just the *expected* return, such zero-shot evaluation is made possible by learning the successor representation (SR; Dayan, 1993). This approach has recently been extended to continuous state spaces (Blier et al., 2021), and the introduction of a variety of density modelling and generative modelling techniques mean that such zero-shot transfer is now possible at scale (Janner et al., 2020; Touati & Ollivier, 2021; Touati et al., 2023).

This paper extends the idea of the successor representation to distributional RL, by defining the *distributional successor measure* (DSM). We show that the DSM is a reward-agnostic object that can by combined with any deterministic reward function to obtain the corresponding distribution of returns, extending zero-shot transfer to the entire distribution of returns. We additionally contribute an algorithm framework for approximating the DSM at scale. Our primary algorithmic contribution is the $\delta$-model, a tractable approximation to the distributional successor measure based on ensembles of diverse generative models. We contribute a training algorithm for $\delta$-models, and several practical implementation techniques that are crucial for success. We exhibit $\delta$-models on several environments, showing that they can be used to obtain meaningful risk-sensitive evaluations of policies against held-out reward functions.

## 2 BACKGROUND

In the sequel, $\mathrm{Law}(X)$ denotes the probability measure governing a random variable $X$, and $X \stackrel{\mathcal{L}}{=} Y$ (read *equal in distribution*) is written to indicate that $\mathrm{Law}(X) = \mathrm{Law}(Y)$. The notation $\mathscr{P}(A)$ defines the space of probability measures over a set $A$. We also write $(X, Y) \sim \mu \otimes \nu$ to refer to the pair of independent samples $X \sim \mu$ and $Y \sim \nu$.

We consider a Markov decision process (MDP) with state space $\mathcal{X}$, finite action space $\mathcal{A}$, transition kernel $p : \mathcal{X} \times \mathcal{A} \to \mathscr{P}(\mathcal{X})$, bounded and measurable reward function $r : \mathcal{X} \to \mathbb{R}$, and discount factor $\gamma \in [0, 1)$. We assume henceforth that $\mathcal{X}$ is a complete and separable metric space, which allows for finite state spaces, as well as many continuous state spaces of interest. Given a policy $\pi : \mathcal{X} \to \mathscr{P}(\mathcal{A})$ and initial state $x_0 \in \mathcal{X}$, an agent generates a random trajectory $(X_t, A_t, R_t)_{t=0}^{\infty}$ of states, actions, and rewards, with distributions specified by $X_0 = x$, $A_t \sim \pi(\cdot|X_t)$, $R_t = r(X_t)$, and $X_{t+1} \sim p(\cdot|X_t, A_t)$ for all $t \geq 0$. For a fixed policy $\pi$, we will denote the transition kernel governing state evolution by $p^\pi$, where $p^\pi(\cdot \mid x) = \sum_{a \in \mathcal{A}} p(\cdot \mid x, a)\pi(a \mid x)$.

The (random) return summarises the performance of the agent along its trajectory, and for each possible initial state $x \in \mathcal{X}$ for the trajectory, it is defined as

$$G_r^\pi(x) := \sum_{t=0}^{\infty} \gamma^t r(X_t), \quad X_0 = x, \ X_{t+1} \sim p^\pi(\cdot \mid X_t). \tag{1}$$

When there is no ambiguity about the reward function, we will write $G^\pi$ in place of $G_r^\pi$. For a given policy $\pi$, the problem of *policy evaluation* is to find the expected return for each initial state. Mathematically, this can be expressed as learning the function $V_r^\pi : \mathcal{X} \to \mathbb{R}$, defined by

$$V_r^\pi(x) := \mathbb{E}\left[G_r^\pi(x)\right]; \tag{2}$$

this describes the quality of $\pi$ in its own right, and may also be used to obtain *improved* policies, for example by acting greedily (Puterman, 2014; Sutton & Barto, 2018).

### 2.1 SUCCESSOR MEASURE

The *normalized successor measure* $\Psi^\pi : \mathcal{X} \to \mathscr{P}(\mathcal{X})$ associated with a policy $\pi$ is defined by

$$\Psi^\pi(S \mid x) := \sum_{t=0}^{\infty} (1 - \gamma)\gamma^t \Pr(X_t \in S \mid X_0 = x), \tag{3}$$

for any (measurable) set $S \subseteq \mathcal{X}$[1] and initial state $x \in \mathcal{X}$. In the literature, $\Psi^\pi(\cdot \mid x)$ is often referred to as the (discounted) state occupancy measure. The object $\Psi^\pi$ described above is a normalised version of the *successor representation* (SR; Dayan, 1993) in the tabular case and the *successor measure* (SM; Blier et al., 2021; Touati & Ollivier, 2021) for continuous state spaces. Blier et al. (2021) shows that, without the $(1 - \gamma)$ factor in Equation 3, $\Psi^\pi(\cdot \mid x)$ is a measure for each $x \in \mathcal{X}$ with total mass $(1 - \gamma)^{-1}$. We include the $(1 - \gamma)$ normalizing factor so that $\Psi^\pi(\cdot \mid x)$ is in fact a probability distribution – this allows for one to sample from the successor measure, as in the work of Janner et al. (2020). Intuitively, $\Psi^\pi(S \mid x)$ describes the proportion of time spent in the region $S \subseteq \mathcal{X}$, in expectation, weighted by the discount factor according to the time of visitation.

Since for each $x \in \mathcal{X}$, $\Psi^\pi(\cdot|x)$ is a probability distribution over states, we can compute expectations with respect to this distribution. Notably, the reward function $r$, successor measure $\Psi^\pi$, and value function $V^\pi$ satisfy the following identity,

$$V_r^\pi(x) = (1 - \gamma)^{-1} \mathbb{E}_{X' \sim \Psi^\pi(\cdot|x)}[r(X')], \tag{4}$$

as leveraged in the recent work of Janner et al. (2020) and Blier et al. (2021). In words, the value function can be expressed as an expectation of the reward, with respect to the successor measure $\Psi^\pi(\cdot|x)$; this expression cleanly factorises the value function into components comprising transition information and reward information, and generalises the result in the tabular case by Dayan (1993). A central consequence is that learning $\Psi^\pi$ allows for the evaluation of $\pi$ on unseen reward functions, without further learning; this is known as *zero-shot policy evaluation*.

---

[1]This covers discounted occupancies over Polish state spaces, including compact Euclidean space.

## 2.2 Distributional policy evaluation

In distributional reinforcement learning (Morimura et al., 2010; Bellemare et al., 2017a; 2023), the problem of *distributional policy evaluation* is concerned with finding not just the expectation of the random return, but its full probability distribution. In analogy with our description of policy evaluation above, this can be mathematically expressed as aiming to learn the return-distribution function $\eta_r^\pi : \mathcal{X} \to \mathscr{P}(\mathbb{R})$, with $\eta_r^\pi(x)$ equal to the distribution of $G_r^\pi(x)$

An added complication in the distributional setting is that the distributional objects to be learned are infinite-dimensional, in contrast with the scalar mean returns learned in classical reinforcement learning. This requires careful consideration of how probability distributions will be represented algorithmically, with common choices including categorical (Bellemare et al., 2017a) and quantile (Dabney et al., 2018b) approaches; see Bellemare et al. (2023, Chapter 5) for a summary.

## 3 Random Occupancy Measures and the Distributional SM

One of the core contributions of this paper is to introduce a mathematical object that plays the role of the successor measure in distributional reinforcement learning. Analogous to how distributional RL models the distribution of the return we propose to study the *distribution* of future state occupancies.

### 3.1 Random occupancy measures

To begin, we contribute a new form for the normalised successor measure (SM), which shows that it can be written as an expectation of the discounted visitation distribution for the *random* state sequence $(X_t)_{t\geq 0}$ generated by $\pi$:

$$\Psi^\pi(S \mid x) = \mathbb{E}_\pi \left[ \sum_{k=0}^\infty (1-\gamma)\gamma^k \delta_{X_k}(S) \,\middle|\, X_0 = x \right] \qquad \forall S \subseteq \mathcal{X} \text{ measurable.} \tag{5}$$

Here, $\delta_{X_k}$ is the probability distribution over $\mathcal{X}$ that puts all its mass on $X_k$, so that $\delta_{X_k}(S) = \mathbb{1}\{X_k \in S\}$. There is now a natural way to obtain a distributional version of this object, by "removing the expectation".

**Definition 1** (Random occupancy measure). *For a given policy $\pi$, let $(X_t)_{t=0}^\infty$ be a random sequence of states generated by interacting with the environment via $\pi$. The associated* random discounted state-occupancy measure $M^\pi$ *assigns to each initial state $x \in \mathcal{X}$ a random probability distribution $M^\pi(\cdot \mid x)$ according to*

$$M^\pi(S \mid x) := \sum_{k=0}^\infty (1-\gamma)\gamma^k \delta_{X_k}(S), \ X_0 = x \quad \forall S \subseteq \mathcal{X} \text{ measurable.} \tag{6}$$

It is worth pausing to consider the nature of the object we have just defined. For each $x \in \mathcal{X}$, $M^\pi(\cdot \mid x)$ is a random variable, and each realisation of $M^\pi(\cdot \mid x)$ is a probability distribution over $\mathcal{X}$. So, for any (measurable) subset of states $Y \subseteq \mathcal{X}$, $M^\pi(Y|x)$ is also a random variable, which gives the discounted proportion of time spent in $Y$ across different possible sampled trajectories. Thus, the distribution of $M^\pi(\cdot \mid x)$ is a distribution *over* probability distributions; see Figure 1.

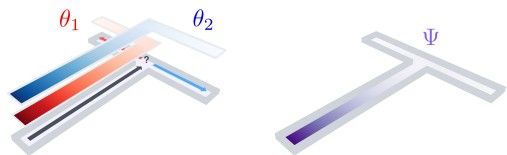

Figure 1: Depiction of the distributional SM in a T-maze, for a policy that chooses a direction at the fork uniformly. **Left:** Atoms of the random occupancy measure. **Right:** The SR, $0.5\theta_1 + 0.5\theta_2$.

As described in Section 2, an important property of the successor representation is that it is a linear operator that maps reward functions to value functions. The next result shows that $M^\pi$ can be used to map reward functions to random returns; all proofs are given in Appendix E.

**Proposition 1.** *Let $M^\pi$ denote a random discounted state-occupancy measure for a given policy $\pi$. For any deterministic reward function $r : \mathcal{X} \to \mathbb{R}$, we have*

$$G_r^\pi(x) \overset{\mathcal{L}}{=} (1-\gamma)^{-1} \mathbb{E}_{X' \sim M^\pi(\cdot|x)} \left[ r(X') \right] . \tag{7}$$

*Note that the right-hand side* is *a random variable, since $M^\pi(\cdot \mid x)$ itself is a random distribution.*

Proposition 1 suggests a novel approach to distributional RL. To obtain return distributions, one can first learn the distribution of $M^\pi$ (without any information about rewards), and then use Equation 7 to obtain an estimate of the corresponding return distribution. This also allows for something not possible in the standard framework of distributional RL: zero-shot distributional policy evaluation. In particular, one can learn the distribution of the random occupancy measure, and then obtain an approximation to the return distribution associated with *any* reward function $r$ without requiring further learning. Once the return distribution obtained, the benefits of distributional RL, such as risk estimation, are immediately available, something not possible using SR in isolation.

**Remark 1.** *Perhaps surprisingly, our assumption of a deterministic reward function made in Proposition 1 is necessary for a linear factorization between reward functions and return distributions. This is due to the statistical dependence between random rewards observed along trajectories and random trajectories themselves. We explore this in more depth in Appendix B; see Proposition 3.*

We observe that the SM is mathematically determined by the one-step transition kernel $P^\pi$; see Proposition 3 in Appendix F for a precise statement and proof of this result. It is therefore the *way* in which the SM represents the transition information of the environment that makes zero-shot policy evaluation possible. The same is true of the distributional SM; it is determined by $P^\pi$ mathematically, but its form makes zero-shot distributional policy evaluation particularly straightforward.

Proposition 1 is the core mathematical insight of the paper; we now seek to develop an algorithmic framework that allows these theoretical ideas to be translated into concrete implementations.

## 3.2 THE DISTRIBUTIONAL SM AND BELLMAN EQUATIONS

Just as in standard distributional RL, where we distinguish between the random return $G^\pi(x)$ and its distribution $\eta^\pi(x)$, we introduce notation that will allow us to express the distribution of $M^\pi(\cdot \mid x)$.

**Definition 2** (Distributional successor measure). *The* distributional successor measure *(distributional SM)* $\daleth^\pi : \mathcal{X} \to \mathscr{P}(\mathscr{P}(\mathcal{X}))$ *is defined by* $\daleth^\pi(x) = \mathrm{Law}(M^\pi(\cdot \mid x))$ *for each* $x \in \mathcal{X}$.

A central result in developing temporal-difference methods for learning $\daleth^\pi$ is to show that $M^\pi$ satisfies a *distributional Bellman equation* (Morimura et al., 2010; Bellemare et al., 2017a).

**Proposition 2.** *Let $M^\pi$ denote the random discounted state-occupancy measure induced by a policy $\pi$. Then $M^\pi$ can be expressed recursively via a distributional Bellman equation, as follows:*

$$M^\pi(S \mid x) \overset{\mathcal{L}}{=} (1-\gamma)\delta_x(S) + \gamma M^\pi(S \mid X') \quad \forall S \subseteq \mathcal{X} \text{ measurable}, \tag{8}$$

*where $X' \sim P^\pi(\cdot \mid x)$, and is independent of $M^\pi$.*

This provides a novel reward-agnostic distributional Bellman equation for random occupancy measures. Note that the multi-dimensional reward distributional Bellman equation studied by Freirich et al. (2019); Zhang et al. (2021b) can be thought of as a special instance of this when $\mathcal{X}$ is finite.

We can also express the distributional SM recursively,

$$\daleth^\pi(x) = \mathbb{E}_{X' \sim p^\pi(\cdot \mid x)} \left[ (\mathrm{b}_{x,\gamma})_\sharp \daleth^\pi(X') \right] \tag{9}$$

where $\mathrm{b}_{x,\gamma} : \mathscr{P}(\mathcal{X}) \to \mathscr{P}(\mathcal{X})$ is given by $\mathrm{b}_{x,\gamma}(\mu) = (1-\gamma)\delta_x + \gamma\mu$ and for $\nu \in \mathscr{P}(\mathscr{P}(\mathcal{X}))$, we have $f_\sharp \nu = \nu \circ f^{-1}$ is the *pushforward* of $\nu$ through $f$ for measurable $f$.

## 4 REPRESENTING AND LEARNING THE DISTRIBUTIONAL SM

The previous section has shown that the distributional SM provides an alternative perspective on distributional reinforcement learning, and opens up possibilities such as zero-shot distributional policy evaluation, not previously possible with existing approaches to distributional RL. However, to turn these mathematical observations into practical algorithms, we need methods for efficiently *representing* and *learning* the distributional SM.

### 4.1 REPRESENTATION BY $\delta$-MODELS

As in standard distributional reinforcement learning, we cannot represent $\daleth^\pi$ within an algorithm exactly, as it is comprised of probability distributions, which are objects having infinitely-many

degrees of freedom. To make matters more complicated still, these are distributions not over the real numbers (as in standard distributional RL), but over $\mathscr{P}(\mathcal{X})$, which may itself have infinitely-many degrees of freedom if $\mathcal{X}$ is infinite. We therefore require a tractable approximate representation. We propose the *equally-weighted particle (EWP) representation*, which is ultimately inspired by the quantile representation of return distributions in standard distributional RL algorithms (Dabney et al., 2018b; Nguyen-Tang et al., 2021). Under this representation, the approximation $\daleth(x)$ to the distributional SM at $x$ is represented as a sum of equally-weighted Dirac masses on the set $\mathscr{P}(\mathcal{X})$:

$$\daleth(x) = \frac{1}{m}\sum_{i=1}^{m}\delta_{\theta_i(x)}\,, \tag{10}$$

with $\theta_i(x) \in \mathscr{P}(\mathcal{X})$ for each $i = 1,\ldots,m$. The approximation problem now reduces to learning appropriate values $((\theta_i(x))_{i=1}^m : x \in \mathcal{X})$ of these Dirac masses.

Since each atom $\theta_i(x)$ is a probability distribution over a potentially large space $\mathcal{X}$, we propose to represent the atoms as *generative models*, in the spirit of $\gamma$-models (Janner et al., 2020). In practice, the generative models can be implemented with function approximators, such as deep neural networks, that take as input noise variables similar to the generator of a generative adversarial network (GAN; Goodfellow et al., 2014). With these choices, we refer to the model in Equation 10 as a $\delta$-*model*; see Figure 2 for an illustration of its various components.

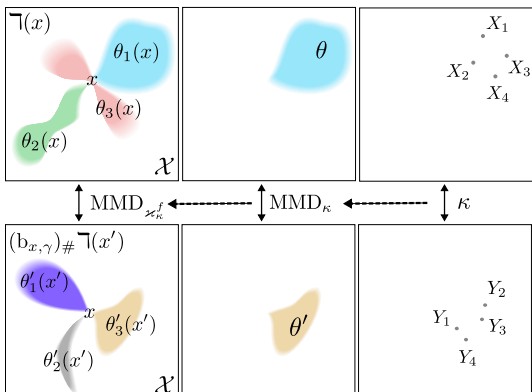

**Terminology.** We have introduced two levels of probability distributions: $\daleth(x)$ itself is a distribution over the generative models $\theta_1(x),\ldots,\theta_m(x)$; and clearly each $\theta_i(x)$ is a distribution over the state space $\mathcal{X}$ itself. To

Figure 2: The components of a $\delta$-model (Section 4.1), and the kernels and distances involved in training them (Section 4.2).

keep track of these two levels, we refer to $\daleth(x)$ as a **model distribution** (that is, a distribution over generative models), and the generative models $\theta_i(x)$ as **state distributions** or **model atoms**. A generative model $\theta \sim \daleth(x)$ distributed according to $\daleth(x)$ is a **model sample**, while a state $X' \sim \theta$ sampled from a generative model is referred to as an **state sample**.

## 4.2 LEARNING FROM SAMPLES

Our goal is to construct an algorithm for learning approximations of the distributions $\daleth^\pi(x)$, parameterized as $\delta$-models, from data. We construct a temporal-difference learning scheme (Sutton, 1984; Dayan, 1993) to approximately solve the distributional Bellman Equation 8 in this metric space, by updating our $\delta$-model $\daleth(x)$ to be closer to the transformation described by the right-hand side of the distributional Bellman equation in Proposition 2, that is

$$\widetilde{\daleth}(x) := \mathbb{E}_{X'\sim p^\pi(\cdot|x)}\left[\frac{1}{m}\sum_{i=1}^{m}\delta_{(1-\gamma)x+\gamma\theta_i(X')}\right]\,. \tag{11}$$

To define an update that achieves this, we will specify a loss function over the space occupied by $\daleth(x)$ (namely $\mathscr{P}(\mathscr{P}(\mathcal{X}))$), distributions *over* distributions of state); this requires care, since this space has such complex structure relative to standard distributional RL problems. We propose to use the *maximum mean discrepancy* (MMDs; Gretton et al., 2012) to construct such a loss. We begin by recalling that for probability distributions $p, q$ over a set $\mathcal{Y}$, the squared-MMD corresponding to the kernel $\kappa : \mathcal{Y} \times \mathcal{Y} \to \mathbb{R}$ is defined as

$$\mathrm{MMD}_\kappa^2(p,q) = \mathbb{E}_{(X,X',Y,Y')\sim p\otimes p\otimes q\otimes q}\left[\kappa(X,X') + \kappa(Y,Y') - 2\kappa(X,Y)\right]\,. \tag{12}$$

**State kernel.** To compare state distributions $\theta, \theta' \in \mathscr{P}(\mathcal{X})$, we will take a **state kernel** $\kappa : \mathcal{X} \times \mathcal{X} \to \mathbb{R}$, and aim to compute $\mathrm{MMD}_\kappa(\theta, \theta')$. Since in $\delta$-models we represent state distributions $\theta, \theta'$ as generative models, we approximate the exact MMD in Equation 12 by instead using samples

from the generative models (Gretton et al., 2012, Eq. 3). If we take $X_1, \ldots, X_{n_1} \overset{\text{i.i.d.}}{\sim} \theta$, and $Y_1, \ldots, Y_{n_2} \overset{\text{i.i.d.}}{\sim} \theta'$ independently, we obtain the following estimator for $\text{MMD}^2_\kappa(\theta_i, \theta_j)$:

$$\widehat{\text{MMD}}^2_\kappa(X_{1:n_1}, Y_{1:n_2}) := \sum_{\substack{i,j=1 \\ i<j}}^{n_1} \frac{\kappa(X_i, X_j)}{\binom{n_1}{2}} + \sum_{\substack{i,j=1 \\ i<j}}^{n_2} \frac{\kappa(Y_i, Y_j)}{\binom{n_2}{2}} - 2 \sum_{i=1}^{n_1} \sum_{j=1}^{n_2} \frac{\kappa(X_i, Y_j)}{n_1 n_2}. \tag{13}$$

**Model kernel.** Equation 13 uses the state kernel to define a metric between generative models. However, ultimately we need a loss function defined at the level of *model distributions* $\daleth(x)$, so that we can define gradient updates that move these quantities towards their corresponding Bellman targets (Equation 11). We now use our notion of distance between state distributions to define a kernel on $\mathscr{P}(\mathcal{X})$ itself, which will allow us to define an MMD over $\mathscr{P}(\mathscr{P}(\mathcal{X}))$, the space of model distributions. To do so, we follow the approach of Christmann & Steinwart (2010, Eq. 6) and Szabo et al. (2015) by defining a **model kernel** $\varkappa_\kappa : \mathscr{P}(\mathcal{X}) \times \mathscr{P}(\mathcal{X}) \to \mathbb{R}$ as a function of $\text{MMD}_\kappa$. In particular, for each $\theta, \theta' \in \mathscr{P}(\mathcal{X})$, we set

$$\varkappa_\kappa(\theta, \theta'; \sigma) = \rho\left(\text{MMD}_\kappa(\theta, \theta')/\sigma\right) \tag{14}$$

for scalar $\sigma > 0$, where $\rho : y \mapsto (1 + y^2)^{-1/2}$ is known as the inverse multiquadric radial basis function. Szabo et al. (2015, Table 1) shows that this kernel is characteristic, and demonstrates other radial basis functions that also yield characteristic kernels.

**DSM MMD loss.** We now specify a loss that will allow us to update $\daleth$ towards the Bellman target in equation 11, by employing the squared MMD under the distribution kernel $\varkappa_\kappa$ defined above:

$$\ell(\daleth, \widetilde{\daleth}; x) = \text{MMD}^2_{\varkappa_\kappa}(\daleth(x), \widetilde{\daleth}(x)). \tag{15}$$

To build a sample-based estimator of this loss, we take a sampled state transition $(x, x')$ generated by the policy $\pi$, and expand the MMD above in terms of evaluations of the kernel $\varkappa_\kappa$; writing $\theta'_i(x') = (1-\gamma)\delta_x + \theta'_i(x)$, this leads to the following loss for the $\delta$-model representation,

$$\frac{1}{m^2} \sum_{i,j=1}^{m} \left( \varkappa_\kappa\Big(\theta_i(x), \theta_j(x)\Big) + \varkappa_\kappa\Big(\theta'_i(x'), \theta'_j(x')\Big) - 2\varkappa_\kappa\Big(\theta_i(x), \theta'_j(x')\Big) \right). \tag{16}$$

Finally, to obtain a loss on which we can compute gradients in practice, each model kernel evaluation above be can be approximated via Equation 14, with the resulting state kernel MMD estimated via Equation 13. Note that we can sample from distributions of the form $(1-\gamma)\delta_x + \gamma\theta_i(x')$ by first sampling $Y \sim \text{Bernoulli}(1-\gamma)$, returning $x$ if $Y = 1$, and otherwise returning an independent sample from $\theta_i(x')$. See Figure 2 for an illustration of how the loss is constructed.

## 5 PRACTICAL TRAINING OF $\delta$-MODELS

In the previous section, we addressed the challenges of representing and learning an approximation of the distributional SM in a computationally tractable manner. This section highlights the implementation of two techniques, namely $n$-step bootstrapping and adaptive kernel design, which we found to be crucial in the optimization of $\delta$-models for the distributional SM. Complete pseudocode for our approach is given in Appendix A.

### 5.1 $n$-STEP BOOTSTRAPPING

The procedure outlined in Section 4.2 computes $\delta$-model targets via one-step bootstrapping. In accordance with Equation 8, the probability mass of the targets due to bootstrapping is $\gamma$, which is generally very high when we are concerned with long horizons. Consequently, particularly at the beginning of training when $\delta$-model estimates are uninformative, the signal-to-noise ratio in the targets is low, which dramatically impedes learning.

Taking inspiration from efforts to reduce the bias of bootstrapping in RL (Watkins, 1989; Sutton & Barto, 2018), we compute $n$-step targets of the distributional SM. By Equation 8, we have

$$M^\pi(\cdot \mid x) \overset{\mathcal{L}}{=} (1-\gamma)\left(\delta_x + \gamma\delta_{X_1} + \gamma^2\delta_{X_2} + \cdots + \gamma^{n-1}\delta_{X_{n-1}}\right) + \gamma^n M^\pi(\cdot \mid X_n), \tag{17}$$

where $X_{k+1} \sim p^\pi(\cdot \mid X_k)$ and $X_0 = x$. An $n$-step version of the DSM MMD loss can then be obtained by replacing the sampled one-step Bellman targets $(1-\gamma)\delta_x + \gamma \daleth(x')$ in Equation 15 with the corresponding $n$-step target $\sum_{k=0}^{n-1}(1-\gamma)\gamma^k\gamma\delta_{x_k} + \gamma^n\theta_i(x_n)$. In analogy with the one-step case, we can sample from this distribution by first sampling $Y$ from a Geometric$(1-\gamma)$ distribution, returning $x_k$ if $Y = k < n$, and returning a sample from $\theta_i(x_n)$ otherwise. Notably, by increasing $n$, we decrease the influence of bootstrap samples on the targets, thereby providing a stronger learning signal grounded in samples from the trajectory.

We found that training stability tends to improve substantially when bootstrap samples account for roughly 80% of the samples in the procedure above. For instance, this approximately corresponds to choosing $n = 5$ when $\gamma = 0.95$. Appendix D includes a more detailed ablation on the choice of $n$. Notably, this procedure for computing TD targets for generative modeling of occupancy measures is not specific to the distributional SM or $\delta$-models. In particular, we suspect this technique would be useful more generally for training geometric horizon models (Janner et al., 2020; Thakoor et al., 2022) with longer horizons, which was reported to be a major challenge (Janner et al., 2020).

## 5.2 Kernel Selection

When training a $\delta$-model with bootstrapped targets, naturally the model/state distributions comprising $\daleth^\pi$ are continually evolving. This poses a challenge when it comes to selecting the kernels we use in practice, since it is not possible *a priori* to decide which kernels can effectively compare distributions on the samples we observe. As such, we found it necessary to employ *adaptive* kernels that evolve with the distributions we are learning.

Powerful methods in the literature involve adversarially learning a kernel over a space of parameterized functions. The MMD-GAN (Li et al., 2017; Binkowski et al., 2018) demonstrates how to parameterize characteristic kernels with deep neural networks. Li et al. (2017) shows that for any characteristic kernel $\kappa : \mathcal{Y} \times \mathcal{Y} \to \mathbb{R}_+$, the function $\kappa \circ f : (x, y) \mapsto \kappa(f(x), f(y))$ is itself a characteristic kernel when $f : \mathcal{X} \to \mathcal{Y}$ is injective. In their work, $f$ is parameterized as the encoder of an autoencoder network, where the autoencoder training encourages $f$ to be injective.

In the case of the distributional SM, parameterizing the model kernel as an injection on the space of probability measures is a major challenge. Rather, we parameterize an adversarial state kernel following the model of Li et al. (2017), using an invertible neural network based on iResNet (Behrmann et al., 2019). Unlike an autoencoder, this *enforces* injectivity, and to our knowledge, no other work has employed invertible neural networks for modeling an adversarial kernel. It should be noted that the state kernel is itself defined as a parameter of the model kernel used in the comparison of $\delta$-models – thus, by adaptively learning the state kernel, our model kernel is itself adaptive.

We also found that further adaptation of the model kernel through the bandwidth $\sigma$ improved training. Our approach is based on the *median heuristic* for bandwidth selection in kernel methods (Takeuchi et al., 2006; Gretton et al., 2012). That is, prior to computing the model MMD, we choose $\sigma^2$ to be the median of the pairwise $\mathrm{MMD}^2_\kappa$ between the models $\theta_i(x)$ of $\daleth(x)$ and those of the bootstrap target $\tilde{\daleth}(x)$, $(1-\gamma)\delta_x + \gamma\theta_j(x')$. Appendix D ablates on our choice of adaptive kernels demonstrating its utility for training $\delta$-models.

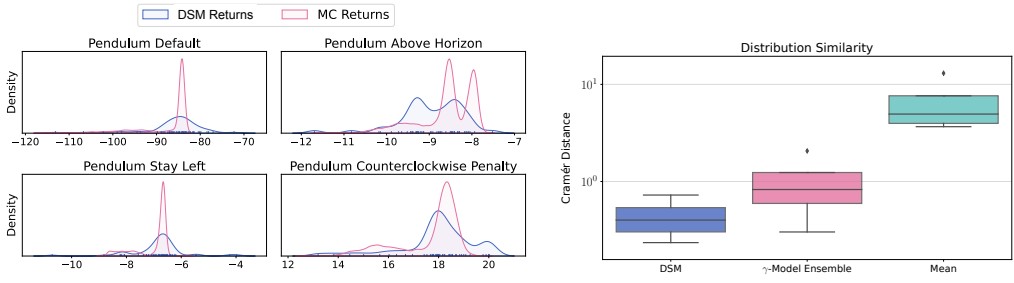

Figure 3: Comparison of return distributions from distributional SM and Monte Carlo.

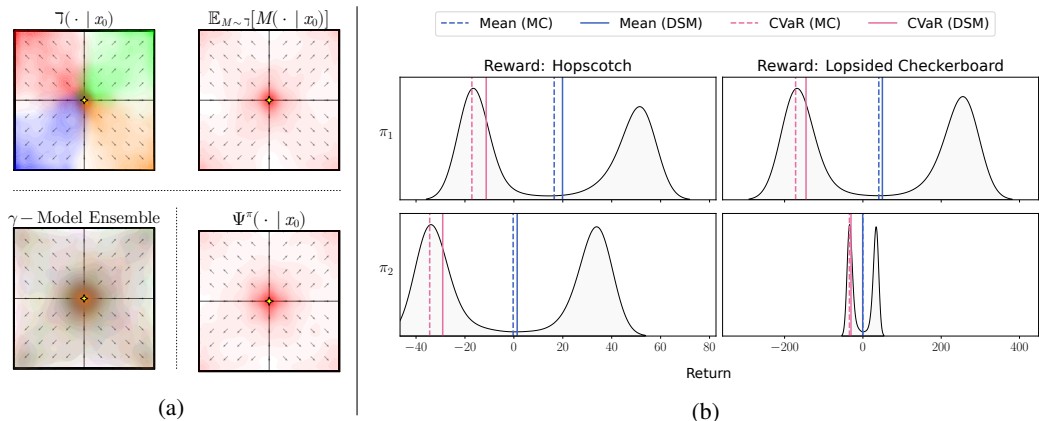

Figure 4: Distributional successor measure predictions in Windy Gridworld. (**4a**): Figures in the left column show the model atoms predicted by the distributional SM (distinguished by color) and by an ensemble of $\gamma$-models. Figures in the right column show the mean over distributional SM model atoms and the SM itself. (**4b**): Distributional SM predictions of return statistics on held-out reward functions for two policies, $\pi_1, \pi_2$. For each reward function, the distributional SM correctly identifies the optimal policy with respect to both mean and CVaR.

## 6 EXPERIMENTAL RESULTS

We evaluate our implementation of the distributional SM on two domains, namely a stochastic "Windy Gridworld" environment, where a pointmass navigates 2D continuous grid subject to random wind force that pushes it towards the corners, and the Pendulum environment (Atkeson & Schaal, 1997). As a baseline, we compare our method to an ensemble of $\gamma$-models (Janner et al., 2020), which approximate the SM with a generative model. We implement the $\gamma$-models with MMD-GAN, similarly to the individual model atoms of a $\delta$-model. We train an ensemble of $m$ $\gamma$-models, where $m$ is the number of models atoms in the comparable $\delta$-model implementation of the distributional SM. Effectively, the ensemble of $\gamma$-models is almost equivalent to a $\delta$-model, with the difference being that the individual $\gamma$-models of the ensemble are trained independently, while the model atoms of a $\delta$-model are coupled through the model MMD loss.

**Visualizing model atoms.** In Figure 4a, we examine the model atoms predicted by our implementation of the $\delta$-model trained on data from a uniform random policy in the Windy Gridworld. Due to the nature of the wind in this domain, which always forces the agent to the corner of the quadrant where it is located, a uniform random policy exhibits a multimodal distribution of model atoms, as shown by the colored densities in the top-left. Alternatively, when examining an ensemble of $\gamma$-models trained on the same data, we see that the models in the ensemble all predict similar state occupancies which align closely with the SM – crucially, only the distributional SM captures the diversity of "futures" that the agent can experience.

**Zero-shot policy evaluation.** A unique feature of the distributional SM is that it acts as an operator that transforms reward functions into return distribution functions. We explored the distributions over returns predicted by the distributional SM for several held-out reward functions and analyzed their similarity with return distributions estimated by Monte Carlo. Figure 3 showcases return distributions predicted by the distributional SM on four tasks in the Pendulum environment meant to model constraints that may be imposed on the system (`Default`, `Above Horizon`, `Stay Left`, `Counterclockwise Penalty`, details in Appendix C.2.2), and it is seen that these predictions capture important statistics, such as the mode and the support of the distributions, which could not be captured by point estimates of the return. Similar results in Windy Gridworld are shown in Appendix C.2.1. For quantitative evaluation, we measure the quality of the return distribution predictions by their dissimilarity to the return distributions estimated by Monte Carlo according to the Cramér distance (Székely & Rizzo, 2013; Bellemare et al., 2017b). We compare the DSM return distribution quality with the quality of return distributions estimated by an ensemble of $\gamma$-models and with Dirac masses centered at the mean of the MC return distributions. Figure 3 demonstrates that the DSM achieves significantly lower Cramér distance than the baselines, indicating that its re-

turn distributions accurately capture aleatoric uncertainty, and provide more statistical information than approaches based on the (expected) SM.

**Risk-sensitive policy selection.** Finally, we demonstrate that distributional SMs can be used to effectively rank policies by various risk-sensitive criteria on held-out reward functions. In Figure 4b, we train distributhional SMs for two different policies, and use them to predict return distributions for two reward functions. We focus on two functionals of these return distributions, namely the mean and the conditional value at risk at level $0.4$ (Rockafellar & Uryasev, 2002, 0.4-CVaR). We see that for both reward functions, the distributional SM accurately estimates both functionals, and is able to correctly identify the superior policy for each criterion. Particularly, for the Lopsided Checkerboard reward, the distributional SM identifies $\pi_1$ as superior with respect to mean reward (identified by locations of solid blue lines), and alternatively identifies $\pi_2$ as superior with respect to $0.4$-CVaR of the return (identified by locations of the solid pink lines). These rankings are validated by the locations of the dashed lines, which are computed by Monte Carlo. We note that, to our knowledge, *no other method can accomplish this feat*. On the one hand, existing distributional RL algorithms could not evaluate the return distributions for held-out reward functions. On the other hand, any algorithm rooted in the SM for zero-shot evaluation can only rank policies by their mean returns, so they must fail to rank $\pi_1, \pi_2$ by at least one of the mean or the $0.4$-CVaR objective.

## 7 RELATED WORK

The successor representation (Dayan, 1993), originally proposed for finite-state MDPs, has been expanded in various directions to handle more general state spaces, beyond the generative-modelling approach to learning the SM described above. Successor features (SFs; Barreto et al., 2017; 2020) model a discounted sum of multi-dimensional state *features*, which can be used for zero-shot evaluation when rewards are expressible as linear functions of the features. A distributional variant of successor features was introduced by Gimelfarb et al. (2021), who additionally modelled component-wise variance of the sum of state features, for entropic risk. This approach also bears a close relationship with the emerging field of multi-variate distributional RL (Freirich et al., 2019; Zhang et al., 2021b); in particular Zhang et al. (2021b) make use of an MMD loss for learning multi-variate return distributions, building on the scalar approach proposed by Nguyen-Tang et al. (2021). Touati & Ollivier (2021) develop a density-modelling approach to learning the SM, in which the optimal policy for any task can be extracted directly, generalizing the universal successor feature approximator of Borsa et al. (2018). Vértes & Sahani (2019) consider the task of learning the SR in *partially observable* MDPs, in particular learning a posterior expectation of the SR defined over latent states.

Beyond transferring knowledge across tasks, learning long-term temporal structure can enhance the representation quality of function approximators for individual sequential decision-making problems (Farebrother et al., 2023; Ghosh et al., 2023), improving exploration (Jinnai et al., 2019; Machado et al., 2020), temporal abstraction (Machado et al., 2018; 2023), and planning (Eysenbach et al., 2021; 2022; Thakoor et al., 2022), as well as other forms of risk-sensitive decision making (Zhang et al., 2021a). The distributional SR also plays a key explanatory role in understanding generalization in RL (Mahadevan & Maggioni, 2007; Stachenfeld et al., 2014; Lan et al., 2022). Additionally, both distributional RL (Dabney et al., 2020; Lowet et al., 2020) and successor representations (Momennejad et al., 2017; Stachenfeld et al., 2014; 2017) have been shown to provide plausible models for biological phenomena in the brain.

## 8 CONCLUSION

This paper presents a fundamentally new approach to distributional RL, which allows for a factorisation of the return distribution into components comprising the immediate reward function, and the *distributional successor measure*. This factorisation reveals the possibility of zero-shot distributional policy evaluation. Notably, this enables efficient comparisons between policies on unseen tasks with respect to arbitrary risk criteria, which no other existing methods have demonstrated. We have also presented a tractable algorithmic framework for training $\delta$-models, which approximate the distributional SM with diverse generative models, and have identified several crucial techniques for large-scale training of $\delta$-models in practice.

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

## A  ALGORITHM

In this section, we restate the core $\delta$-model update derived in Section 4, including the $n$-step bootstrapping and adversarial kernel modifications described in Section 5. Interested practitioners are advised to consult EINOPS (Rogozhnikov, 2022).

---

**Algorithm 1** $\delta$-model update.

---

**Require:** Policy $\pi$ with stationary distribution $d_\pi$, GAN generator $\Phi$, GAN parameters $\{\zeta_i\}_{i=1}^m$ and target parameters $\{\bar{\zeta}_i\}_{i=1}^m$, discriminator function $f$, adversarial kernel parameters $\{\xi_i\}_{i=1}^m$, kernel $\kappa$, step sizes $\alpha, \lambda$, number of state samples $s$.

**while** training **do**
  ▷ Discriminator (adversarial kernel): maximize model MMD
  Set $\xi_i \leftarrow \xi_i + \alpha \nabla_{\xi_i} \ell(\{\zeta_j\}_{j=1}^m, \{\bar{\zeta}_j\}_{j=1}^m, \{\xi_j\}_{j=1}^m)$ for $i = 1, \ldots, m$
  ▷ Generator: minimize model MMD
  Set $\zeta_i \leftarrow \xi_i - \alpha \nabla_{\zeta_i} \ell(\{\zeta_j\}_{j=1}^m, \{\bar{\zeta}_j\}_{j=1}^m, \{\xi_j\}_{j=1}^m)$ for $i = 1, \ldots, m$
  ▷ Generator: target parameter update
  Set $\bar{\zeta}_i \leftarrow (1 - \lambda)\bar{\zeta}_i + \lambda \zeta_i$ for $i = 1, \ldots, m$
**end while**

**procedure** $\ell(\{\zeta_i\}_{i=1}^m, \{\bar{\zeta}_i\}_{i=1}^m, \{\xi_i\}_{i=1}^m)$
  Sample $x_1 \sim d_\pi, x_k \sim P^\pi(\cdot \mid x_{k-1})$ for $k = 2, \ldots, n$.
  **for** $i = 1, \ldots, m$ **do**
    Sample $z_i^1, \ldots, z_i^s$ i.i.d. from GAN noise distribution
    Set $x_i^j \leftarrow \Phi(z_i^j; x_1, \zeta_i)$ for $j = 1, \ldots, s$
    Sample $\omega_i^1, \ldots, \omega_i^s$ i.i.d. from GAN noise distribution
      and $Y_i^1, \ldots, Y_i^s$ i.i.d. from Geometric$(1 - \gamma)$
    Set $\bar{x}_i^j \leftarrow \Phi(\omega_i^j; x_n, \bar{\zeta}_i)$ if $Y_i^j \geq n$, else set $\bar{x}_i^j \leftarrow x_{Y_i^j}$, for $j = 1, \ldots, s$
    Set $y_i^j \leftarrow f(x_i^j, \xi_i)$ for $j = 1, \ldots s$       ▷ Adversarial Kernel Transformations
    Set $\bar{y}_i^j \leftarrow f(\bar{x}_i^j, \xi_i)$ for $j = 1, \ldots s$
  **end for**
  **for** $i = 1, \ldots, m$ **do**       ▷ MMDs Between Source Model Atoms
    **for** $i' = 1, \ldots, m$ **do**

$$\text{Set } d_{i,i'}^s \leftarrow \frac{1}{\binom{s}{2}} \sum_{\substack{l,k=1 \\ l<k}}^s \kappa(y_i^k, y_i^l) + \frac{1}{\binom{s}{2}} \sum_{\substack{l,k=1 \\ l<k}}^s \kappa(y_{i'}^k, y_{i'}^l) - \frac{2}{s^2} \sum_{l,k=1}^s \kappa(y_i^k, y_{i'}^l) \qquad \text{▷ Equation 13}$$

    **end for**
  **end for**
  **for** $i = 1, \ldots, m$ **do**       ▷ MMDs Between Target Model Atoms
    **for** $i' = 1, \ldots, m$ **do**

$$\text{Set } d_{i,i'}^t \leftarrow \frac{1}{\binom{s}{2}} \sum_{\substack{l,k=1 \\ l<k}}^s \kappa(\bar{y}_i^k, \bar{y}_i^l) + \frac{1}{\binom{s}{2}} \sum_{\substack{l,k=1 \\ l<k}}^s \kappa(\bar{y}_{i'}^k, \bar{y}_{i'}^l) - \frac{2}{s^2} \sum_{l,k=1}^s \kappa(\bar{y}_i^k, \bar{y}_{i'}^l)$$

    **end for**
  **end for**
  **for** $i = 1, \ldots, m$ **do**       ▷ MMDs Across Source and Target Model Atoms
    **for** $i' = 1, \ldots, m$ **do**

$$\text{Set } d_{i,i'}^{st} \leftarrow \frac{1}{s^2} \sum_{l,k=1}^s \kappa(y_i^k, \bar{y}_i^l) + \frac{1}{s^2} \sum_{l,k=1}^s \kappa(y_{i'}^k, \bar{y}_{i'}^l) - \frac{2}{s^2} \sum_{l,k=1}^s \kappa(y_i^k, \bar{y}_{i'}^l)$$

    **end for**
  **end for**
  Set $\sigma^2 = \text{Median}\left(\text{Concat}\left(\{d_{i,i'}^s\}, \{d_{i,i'}^t\}, \{d_{i,i'}^{st}\}\right)\right)$     ▷ Adaptive Model Kernel Bandwidth

  Set $L \leftarrow \frac{1}{m^2} \sum_{i,j=1}^m \left( \rho(\sqrt{d_{i,j}^s / \sigma^2}) + \rho(\sqrt{d_{i,j}^t / \sigma^2}) - 2\rho(\sqrt{d_{i,j}^{st} / \sigma^2}) \right)$     ▷ Model MMD

**end procedure**

---

## B FURTHER DISCUSSION AND EXTENSIONS

### B.1 EXAMPLES OF DISTRIBUTIONAL SMS IN FINITE-STATE-SPACE ENVIRONMENTS

In this section, we include several examples to illustrate the breadth of distributions on the simplex that can be obtained as distributional SMs for simple environments.

Figure 5 illustrates a kernel density approximation to the distributional SM in a three-state MDP, with state-transition kernel given by

$$\begin{pmatrix} 0.5 & 0.5 & 0 \\ 0 & 0 & 1 \\ 1/3 & 1/3 & 1.3 \end{pmatrix},$$

and a discount factor of $\gamma = 0.7$. The figure is specifically created by generating 1,000 trajectories of length 100, which are then converted into visitation distributions, serving as approximate samples of the distributional SM, and a kernel density estimator (KDE) is then fitted; we use Seaborn's `kdeplot` method with default parameters (Waskom, 2021). Also included in the figure are corresponding return distribution estimates, obtained by using the identity in Equation 7 with the generated samples described above, and again using a KDE plot of the resulting return distribution estimator. Observe that since the second state transitions deterministically into the third state, the distributional SM for the second state is a scaling and translation of the distributional SM of the third state, as predicted by the distributional SM Bellman equation in Equation 8.

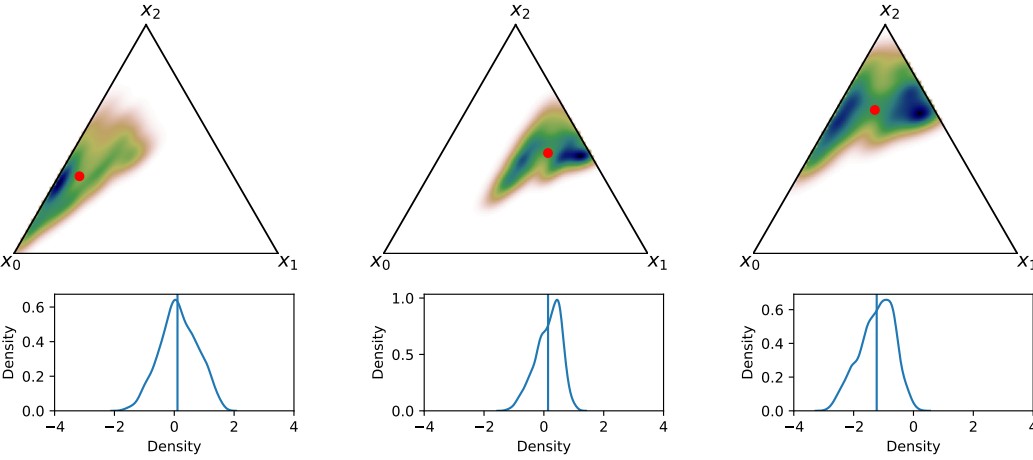

Figure 5: **Top:** Kernel density estimate of distributional SM. Red dot represents the standard SR. **Bottom:** Kernel density estimates of return distributions, obtained via distributional SM. Vertical lines represent expected return, obtained from standard SR.

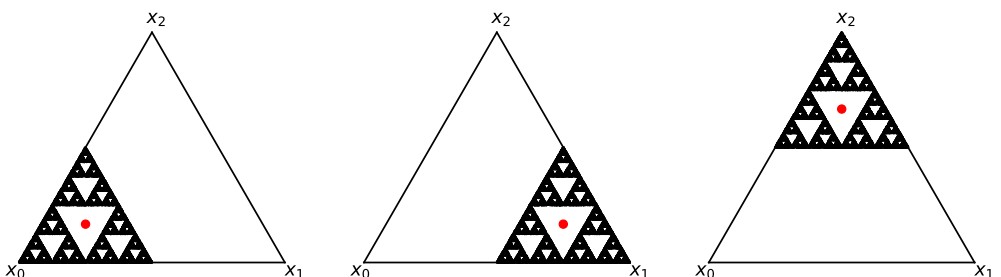

Figure 6: Monte Carlo estimation of the distributional SM at states $x_0$, $x_1$, and $x_2$, in a three-state MDP. Each distribution is supported on a copy of the fractal Sierpiński triangle. Red dot represents the standard SR.

In Figure 6, we plot a Monte Carlo approximation to the distributional SM in a three-state environment in which there is an equal probability of jumping to each state in every transition, and the discount factor is $\gamma = 0.5$. The distributions over the simplex in this case are instances of the Sierpiński triangle, a fractal distribution that is neither discrete nor absolutely continuous with respect to Lebesgue measure on the simplex. This can be viewed as a higher-dimensional analogue of the Cantor distribution described in the context of distributional reinforcement learning in (Bellemare et al., 2023, Example 2.11). These plots were generated using 10,000 samples per state, with an episode length of 100.

### B.2    STOCHASTIC REWARD FUNCTIONS

In the main paper, we make a running assumption that the rewards encountered at each state are given by a deterministic assumption. In full generality, Markov decision processes allow for the state-conditioned reward to follow a non-trivial probability distribution. In this section, we briefly describe the main issue with extending our approach to dealing with stochastic rewards.

The issue stems from the fact that the mapping from sequences of state $(X_k)_{k \geq 0}$ to the corresponding occupancy distribution $\sum_{k=0}^{\infty} \gamma^k \delta_{X_k}$ is often not injective. To see why, consider an environment with four states $x_0, x_1, x_2, x_3$ (including a terminal state $x_3$, which always transitions to itself). Consider two state sequences:

$$(x_0, x_1, x_2, x_2, x_3, x_3, \dots),$$
$$(x_0, x_2, x_1, x_1, x_3, x_3, \dots).$$

These sequences give rise to the visitation distributions

$$(1-\gamma)\delta_{x_0} + (1-\gamma)\gamma\delta_{x_1} + (1-\gamma)(\gamma^2 + \gamma^3)\delta_{x_2} + \gamma^4 \delta_{x_3},$$
$$(1-\gamma)\delta_{x_0} + (1-\gamma)(\gamma^2 + \gamma^3)\delta_{x_1} + (1-\gamma)\gamma\delta_{x_2} + \gamma^4 \delta_{x_3}.$$

Now suppose $\gamma = \gamma^2 + \gamma^3$; clearly there is a value of $\gamma \in (0,1)$ satisfying this equation. But for this value of $\gamma$, the two visitation distributions above are identical. In the case of deterministic state-conditioned rewards, the two corresponding returns are also identical in this case. However, in the case of non-deterministic returns, the corresponding distributions over return are distinct. To give a concrete case, consider the setting in which all rewards are deterministically 0, except at state $x_1$, where they are given by the $\mathrm{N}(0,1)$ distribution. Then under the first visitation distribution, the corresponding return distribution is the distribution of $\gamma Z$ (where $Z \sim \mathrm{N}(0,1)$), which has distribution $\mathrm{N}(0, \gamma^2)$. In contrast, the return distribution for the second visitation distribution is the distribution of $\gamma^2 Z + \gamma^3 Z'$ (where $Z, Z' \overset{\text{i.i.d.}}{\sim} N(0,1)$), which has distribution $N(0, \gamma^4 + \gamma^6)$. However, $\gamma^2 \neq \gamma^4 + \gamma^6$, and hence these distributions are not equal.

These observations mean that the framework *can* be extended to handle stochastic rewards in cycle-less environments; that is, environments where each state can be visited at most once in a given trajectory. This incorporates the important class of finite-horizon environments.

### B.3    THE SUCCESSOR MEASURE AS A LINEAR OPERATOR

Here, we recall a key notion from Blier et al. (2021) used in several proofs that follow. Successor measures act naturally as linear operators on the space $\mathrm{B}(\mathcal{X})$ of bounded measurable functions, much in the same way as Markov kernels act as linear operators (see e.g. Le Gall (2016)). Particularly, for any $f \in \mathrm{B}(\mathcal{X})$, we write

$$(\Psi^\pi f)(x) = \int_{\mathcal{X}} f(x') \Psi^\pi(\mathrm{d}x' \mid x), \tag{18}$$

noting that $\Psi^\pi(\cdot \mid x)$ is a (probability) measure for each $x \in \mathcal{X}$. Through this linear operation, the successor measure transforms reward functions $r : \mathcal{X} \to \mathbb{R}$ to value functions $V_r^\pi$,

$$(1-\gamma)V_r^\pi(x) = \mathbb{E}_\pi \left[ \sum_{t \geq 0} (1-\gamma)\gamma^t r(X_t) \mid X_0 = x \right]$$
$$= \mathbb{E}_{X' \sim \Psi^\pi(\cdot \mid x)}[r(X_t)]$$
$$= (\Psi^\pi r)(x).$$

## C  EXPERIMENTAL DETAILS

In this section, we provide additional details relating to the experiments in the main paper.

### C.1  HYPERPARAMETERS

Unless otherwise specified the default hyperparameters used for our implementation of $\delta$-model are outlined in Table 1. Certain environment specific hyperparameters can be found in Appendix C.2.

Table 1: Default hyperparameters for $\delta$-model.

| Hyperparameter | Value |
|---|---|
| Generator Network | MLP(3-layers, 256 units, ReLU) |
| Generator Optimizer | Adam($\beta_1 = 0.9$, $\beta_2 = 0.999$) |
| Generator Learning Rate | $6.25e-5$ |
| Discriminator Network | iResMLP(2 layers $\times$ 2 blocks, 256 units, ReLU) |
| Discriminator Optimizer | Adam($\beta_1 = 0.9$, $\beta_2 = 0.999$) |
| Discriminator Learning Rate | $6.25e-5$ |
| Discriminator Feature Dimensionality | 8 output features |
| Model Kernel | InverseMultiQuadric |
| Adaptive Model Kernel (Median Heuristic) | True |
| State Kernel | RationalQuadricKernel($\mathcal{A} = \{0.2, 0.5, 1.0, 2.0, 5.0\}$) |
| Adaptive State Kernel (Adversarial Kernel) | True |
| Horizon ($n$-step) | 5 |
| Discount Factor ($\gamma$) | 0.95 |
| Batch Size | 32 |
| Number of State Samples | 32 |
| Number of Model Samples | 51 |
| Target Parameter Step Size ($\lambda$) | 0.01 |
| Noise Distribution | $\omega \in \mathbb{R}^8 \sim \mathcal{N}(0, I)$ |
| Number of Gradient Updates | $3e6$ |

### C.2  ENVIRONMENT DETAILS

Below we provide specifics of the environments utilized for the experimental results in the paper.

#### C.2.1  WINDY GRIDWORLD

When training a $\delta$-model for the Windy Gridworld experiments, we use 4 model atoms and train for 1 million gradient steps.

Our experiments in Section 5 involve two reward functions, namely `Hopscotch` and `Lopsided Checkerboard`. These reward functions have constant rewards in each quadrant, as shown in Figure 7.

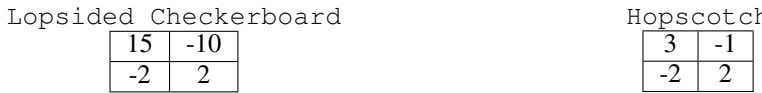

Figure 7: Reward functions for Windy Gridworld.

Moreover, we provide some additional visualizations on predicted return distributions from our distributional SM implementation in Figure 8.

Notably, Figure 8 demonstrates that the distributional SM correctly predict the fraction of futures that enter the red region, which demonstrates that the distributional SM is can detect when a policy will be likely to violate novel constraints.

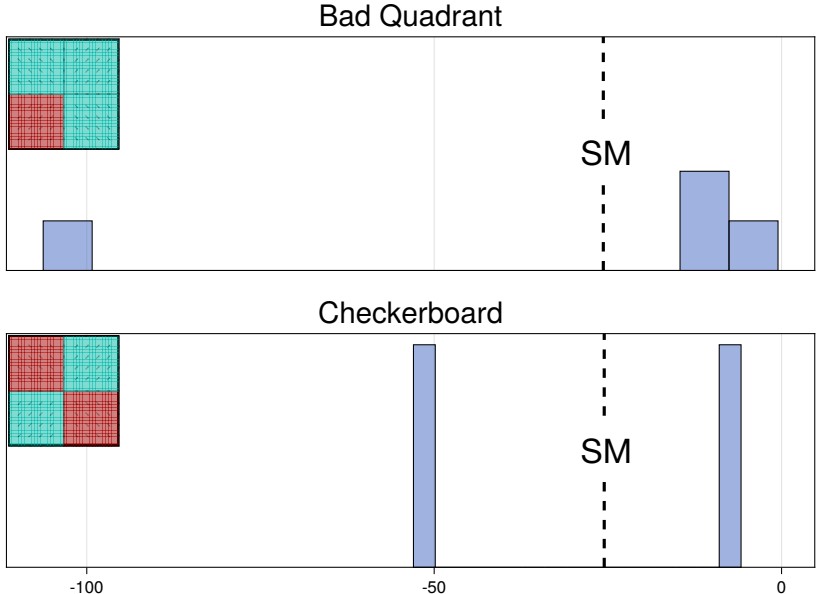

Figure 8: Return distribution predictions in Windy Gridworld under the uniform random policy where the source state is the origin. Each row represents a separate reward function depicted by the inset grids, with red regions denoting negative reward.

### C.2.2 PENDULUM

When training a $\delta$-model for the Pendulum experiments, we use 51 model atoms and train for 3 million gradient steps.

Our experiments on the Pendulum environment involve zero-shot policy evaluation for rewards that are held out during training. We considered four reward functions, namely `Default`, `Above Horizon`, `Stay Left`, and `Counterclockwise Penalty`, which we describe below.

All reward functions are defined in terms of the pendulum angle $\theta \in [-\pi, \pi]$, its angular velocity $\dot{\theta}$, and the action $a \in \mathbb{R}$. The reward functions are given by

$$r_{\texttt{Default}}(\theta, \dot{\theta}, a) = -\left(\theta^2 + 0.1\dot{\theta}^2 + 0.001a^2\right)$$

$$r_{\texttt{Above Horizon}}(\theta, \dot{\theta}, a) = -(\mathbb{1}_{\theta \geq \pi/2} + 0.1a^2)$$

$$r_{\texttt{Stay Left}}(\theta, \dot{\theta}, a) = \min(0, \sin\theta)$$

$$r_{\texttt{CCW Penalty}}(\theta, \dot{\theta}, a) = \mathbb{1}_{\dot{\theta} < 0}$$

These reward functions (aside from `Default`) were chosen to model potential constraints that can be imposed on the system after a learning phase. The `Above Horizon` reward imposes extra penalty whenever the pendulum is below the horizon, which may model the presence of an obstacle under the horizon. The `Stay Left` reward reinforces the system when the pendulum points further to the left, which could, for instance, indicate a different desired target for the pendulum. Finally, the `Counterclockwise Penalty` ($r_{\texttt{CCW Penalty}}$ above) reinforces the system for rotating clockwise, which can model a constraint on the motor.

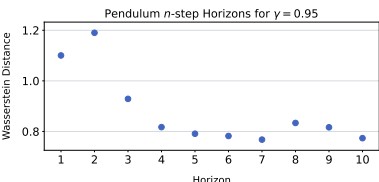 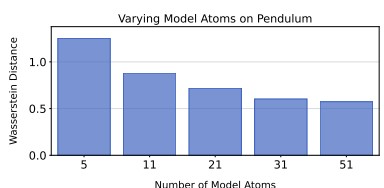 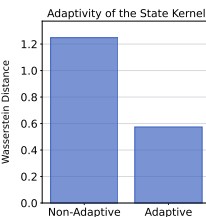

Figure 9: Each subplot indicates the Wasserstein distance of the DSM versus the empirical MC return distributions on 9 source states for 4 reward functions on Pendulum. **Left:** Varying the value of $n$ in the $\delta$-model multi-step bootstrapped target. **Middle:** varying the number of model atoms. **Right:** Selectively applying an adaptive (adversarial) or non-adaptive state kernel.

## D    ABLATION EXPERIMENTS

As mentioned in Section 5 there are several practical considerations when learning $\delta$-models. We further expand on three crucial details: $n$-step bootstrapping, adaptive kernels, and how the number of model atoms affects our approximation error.

$n$**-step bootstrapping.** In order to allow $\delta$-models to learn longer horizons than typically possible with $\gamma$-models (Janner et al., 2020) we employ the use of $n$-step bootstrapping when constructing our target distribution. The choice of $n$ is critical; if $n$ is too small, the update is over-reliant on bootstrapping, leading to instability. Conversely, for large $n$ it becomes impractical to store long sequences.

For these reasons it is worth understanding how $\delta$-models interacts with the value of $n$, which can help guide the selection of this parameter for any type of geometric horizon model (e.g., Janner et al., 2020; Thakoor et al., 2022). To this end, we perform a sweep over $n \in \{1, 2, \ldots, 10\}$ to understand how this affects the Wasserstein distance between the DSM-estimated return compared to the empirical MC return distribution. We train a $\delta$-model with $\gamma = 0.95$ for each $n$ on the Pendulum environment with all other hyperparameters remaining fixed as in Appendix C.1. Figure 9 (left) shows the Wasserstein distance averaged over 9 source states for the four reward functions outlined in Appendix C.2.2.

We can see that $n$-step bootstrapping does indeed help us to learn better approximations until around $n = 5$ where the benefits are less clear. This corresponding to the bootstrapped term accounting for $\approx 80\%$ of the probability mass of the target distribution.

**Adversarial kernel.** Given the non-stationary of our target distributions, we found it crucial to employ an adaptive kernel in the form of an adversarial kernel (Li et al., 2017) for the state kernel. Note that the model kernel is itself a function of the state kernel so by learning an adversarial state kernel we are able to adapt our model kernel as well.

To validate our decision to employ an adversarial kernel we train a $\delta$-model with and without an adaptive kernel. The adaptive kernel is the one described in Appendix C.1. The non-adaptive kernel omits the application of the learned embedding network, that is, the kernel is a mixture of rational quadric kernels,

$$\rho(d) = \sum_{\alpha \in \mathcal{A}} \left( 1 + \frac{d}{2\alpha} \right)^{-\alpha},$$

for $\mathcal{A} = \{0.2, 0.5, 1.0, 2.0, 5.0\}$ as per Binkowski et al. (2018). Figure 9 (right) shows that the Wasserstein distance is nearly halved when applying the adversarial kernel.

**Number of model atoms.** As the number of model atoms increases we expect to better approximate $\daleth^\pi$. To get an idea of how our approximation is improving as we scale the number of atoms we compare the Wasserstein distance from our $\delta$-model to the empirical MC return distributions for $\{5, 11, 21, 31, 41, 51\}$ model atoms. These results are presented in Figure 9 (Middle). As expected we obtain a better approximation to $\daleth^\pi$ when increasing the number of model atoms. Further scaling the number of model atoms should continue to improve performance at the cost of compute. This is a desirable property for risk-sensitive applications where better approximations are required.

## E  Proofs

**Proposition 1.** *Let $M^\pi$ denote a random discounted state-occupancy measure for a given policy $\pi$. For any deterministic reward function $r : \mathcal{X} \to \mathbb{R}$, we have*

$$G_r^\pi(x) \stackrel{\mathcal{L}}{=} (1-\gamma)^{-1} \mathbb{E}_{X' \sim M^\pi(\cdot|x)} \left[ r(X') \right] . \tag{7}$$

*Note that the right-hand side* is *a random variable, since $M^\pi(\cdot \mid x)$ itself is a random distribution.*

*Proof.* This result can be verified by a direct calculation. Invoking Definition 1, we have

$$
\begin{aligned}
(M^\pi r)(x) &= \int_{\mathcal{X}} r(x') M^\pi(\mathrm{d}x' \mid x) \\
&\stackrel{\mathcal{L}}{=} \int_{\mathcal{X}} \sum_{t \geq 0} (1-\gamma)\gamma^t r(x') \delta_{X_t}(\mathrm{d}x') \ \bigg| \ X_0 = x \\
&\stackrel{\mathcal{L}}{=} (1-\gamma) \sum_{t \geq 0} \gamma^t r(X_t) \ \bigg| \ X_0 = x \\
&\stackrel{\mathcal{L}}{=} (1-\gamma) G_r^\pi(x)
\end{aligned}
$$

where the third step invokes Fubini's theorem subject to the boundedness of $r$. The claimed result simply follows by dividing through by $1 - \gamma$. $\qquad \square$

**Proposition 2.** *Let $M^\pi$ denote the random discounted state-occupancy measure induced by a policy $\pi$. Then $M^\pi$ can be expressed recursively via a distributional Bellman equation, as follows:*

$$M^\pi(S \mid x) \stackrel{\mathcal{L}}{=} (1-\gamma)\delta_x(S) + \gamma M^\pi(S \mid X') \quad \forall S \subseteq \mathcal{X} \text{ measurable}, \tag{8}$$

*where $X' \sim P^\pi(\cdot \mid x)$, and is independent of $M^\pi$.*

*Proof.* By Definition 1, we have

$$
\begin{aligned}
M^\pi(\cdot \mid x) &\stackrel{\mathcal{L}}{=} (1-\gamma)\delta_x + \sum_{t=1}^{\infty} (1-\gamma)\gamma^t \delta_{X_t} \\
&\stackrel{\mathcal{L}}{=} (1-\gamma)\delta_x + \gamma \sum_{t=0}^{\infty} (1-\gamma)\gamma^t \delta_{X_{t+1}} \\
&\stackrel{\mathcal{L}}{=} (1-\gamma)\delta_x + \gamma M^\pi(\cdot \mid X') ,
\end{aligned}
$$

with the final equality in distribution following from the Markov property. $\qquad \square$

## F  Additional results

**Proposition 3.** *The distributional SM is determined by the standard SR. In other words, given $\Psi^\pi$, one can mathematically derive $\daleth^\pi$.*

*Proof.* To establish Proposition 3, it suffices to show that the one-step transition kernel $p^\pi$ for a given policy $\pi$ can be recovered exactly from $\Psi^\pi$. This is because $p^\pi$ contains all possible structural information about the environment and the policy's dynamics, so it contains all information necessary to construct the distributional SM. When $\mathcal{X}$ is finite, Lemma 1 shows that $\Psi^\pi$ encode $p^\pi$, and Lemma 2 demonstrates this for the more general class of state space $\mathcal{X}$ considered in this paper. $\quad \square$

**Lemma 1.** *Let $\mathcal{X}$ be finite, and let $\Psi^\pi$ denote the successor representation for a given policy $\pi$. Then $p^\pi$ can be recovered exactly from $\Psi^\pi$.*

*Proof.* Consider a policy $\pi : \mathcal{X} \to \mathscr{P}(\mathcal{A})$ with discounted visitation distributions $\Psi^\pi$. We consider the state transition matrix $P^\pi \in \mathbb{R}^{|\mathcal{X}| \times |\mathcal{X}|}$ where $P^\pi_{x,x'} = p^\pi(x' \mid x)$. Recall that $\Psi^\pi = (1 - \gamma)(I - \gamma P^\pi)^{-1}$, so rearranging we have $P^\pi = \gamma^{-1}(I - (1 - \gamma)(\Psi^\pi)^{-1})$. Therefore the one-step state-to-state transition probabilities are determined by $\Psi^\pi$, and since $\daleth^\pi$ is a function of of the one-step transition probabilities, the conclusion follows. $\qquad\square$

**Lemma 2.** *Let $\mathcal{X}$ be a complete, separable metric space endowed with its Borel $\sigma$-field $\Sigma$, and let $\Psi^\pi$ denote the successor measure for a given policy $\pi$. Then $\Psi^\pi$ encodes $p^\pi$, in the sense that $p^\pi$ can be expressed as a function of $\Psi^\pi$ alone.*

*Proof.* Recall the definition of the successor measure $\Psi^\pi : \Sigma \to \mathbb{R}^+$,

$$\Psi^\pi(A \mid x) = (1 - \gamma) \sum_{t \geq 0} \gamma^t \Pr(X_t \in A \mid X_0 = x).$$

As shown above in Appendix B.3, $\Psi^\pi$ acts as a linear operator on $\mathrm{B}(\mathcal{X})$ according to $(\Psi^\pi f)(x) = \mathbb{E}_{X' \sim \Psi^\pi(\cdot \mid x)}[f(X')]$. We denote by $P^\pi : \mathrm{B}(\mathcal{X}) \to \mathrm{B}(\mathcal{X})$ the Markov kernel corresponding to $p^\pi$, where $\mathrm{B}(\mathcal{X})$ denotes the space of bounded and measurable functions on $\mathcal{X}$. The operator $P^\pi$ acts on a function $f \in \mathrm{B}(\mathcal{X})$ according to

$$(P^\pi f)(x) = \int_{\mathcal{X}} f(x') p^\pi(\mathrm{d}x' \mid x) = \mathbb{E}_{X' \sim p^\pi(\cdot \mid x)}[f(X')].$$

That is, $(P^\pi f)(x)$ computes the expected value of $f$ over the distribution of next states, conditioned on a starting state. Returning to the definition of the successor measure, for any $f \in \mathrm{B}(\mathcal{X})$, we have

$$(\Psi^\pi f)(x) = \int_{\mathcal{X}} f(x') \Psi^\pi(\mathrm{d}x' \mid x)$$

$$= (1 - \gamma) \int_{\mathcal{X}} f(x') \sum_{t \geq 0} \gamma^t (p^\pi)^t (\mathrm{d}x' \mid x)$$

$$= (1 - \gamma) \sum_{t \geq 0} \gamma^t \int_{\mathcal{X}} f(x') (p^\pi)^t (\mathrm{d}x' \mid x)$$

$$= (1 - \gamma) \sum_{t \geq 0} \gamma^t ((P^\pi)^t f)(x)$$

where the third step invokes Fubini's theorem, given the boundedness of $f$ and $p^\pi$. We have shown that

$$\Psi^\pi = (1 - \gamma) \sum_{t \geq 0} \gamma^t (P^\pi)^t,$$

where the correspondence is with respect to the interpretation of $\Psi^\pi$ as a linear operator on $\mathrm{B}(\mathcal{X})$. Blier et al. (2021, Theorem 2) show that $\sum_{t \geq 0} \gamma^t (P^\pi)^t = (\mathrm{id} - \gamma P^\pi)^{-1}$ as linear operators on $\mathrm{B}(\mathcal{X})$, where id is the identity map on $\mathrm{B}(\mathcal{X})$. As a consequence, $\Psi^\pi$ is proportional to the inverse of a linear operator, so it is itself an invertible linear operator, where

$$(\Psi^\pi)^{-1} = \frac{1}{1 - \gamma}(\mathrm{id} - \gamma P^\pi)$$

and hence

$$P^\pi = \gamma^{-1} \left( \mathrm{id} - (1 - \gamma)(\Psi^\pi)^{-1} \right).$$

Again, the correspondence is established for $P^\pi$ as a linear operator on $\mathrm{B}(\mathcal{X})$. However, we can now recover the measures $p^\pi(\cdot \mid x)$ according to

$$p^\pi(A \mid x) = \int_A p^\pi(\mathrm{d}x' \mid x)$$

$$= \int_{\mathcal{X}} \chi_A(x') p^\pi(\mathrm{d}x' \mid x) \qquad\qquad (\chi_A(y) \triangleq \mathbb{1}_{y \in A})$$

$$= (P^\pi \chi_A)(x)$$

$$= \left( \gamma^{-1}(\mathrm{id} - (1 - \gamma)(\Psi^\pi)^{-1}) \chi_A \right)(x)$$

where $\chi_A \in \mathrm{B}(\mathcal{X})$ for any measurable set $A$. Thus, we have shown that $p^\pi(\cdot \mid x)$ can be reconstructed from $\Psi^\pi$ alone, as claimed. $\qquad\square$

### F.1 DISTRIBUTIONAL DYNAMIC PROGRAMMING

In this section, we demonstrate how the distributional SM can be computed by dynamic programming. Following familiar techniques in the analysis of dynamic programming algorithms, we will demonstrate that the distributional SM is the unique fixed point of a contractive operator, and appeal to the Banach fixed point theorem.

To begin, we will define the operator of interest, which we refer to as the distributional Bellman operator $\mathcal{T}^\pi : \mathscr{P}(\mathscr{P}(\mathcal{X}))^{\mathcal{X}} \to \mathscr{P}(\mathscr{P}(\mathcal{X}))^{\mathcal{X}}$,

$$(\mathcal{T}^\pi \daleth)(x) = \mathbb{E}_{X' \sim p^\pi(\cdot|x)} \left[ (\mathrm{b}_{x,\gamma})_\sharp \daleth(X') \right].$$

It follows directly from equation 9 that $\daleth^\pi = \mathcal{T}^\pi \daleth^\pi$.

**Proposition 4.** *Let $d$ be a metric on $\mathcal{X}$ such that $(\mathcal{X}, d)$ is a Polish space, and let $w_d$ denote the Wasserstein distance on $\mathscr{P}(\mathcal{X})$ with base distance $d$. Then, if $W : \mathscr{P}(\mathscr{P}(\mathcal{X})) \times \mathscr{P}(\mathscr{P}(\mathcal{X})) \to \mathbb{R}$ is the Wasserstein distance on $\mathscr{P}(\mathscr{P}(\mathcal{X}))$ with base distance $w_d$, we have*

$$\overline{W}(\mathcal{T}^\pi \daleth_1, \mathcal{T}^\pi \daleth_2) \leq \gamma \overline{W}(\daleth_1, \daleth_2),$$

*where $\overline{W}$ is the "supremal" $W$ metric given by $\overline{W}(\daleth_1, \daleth_2) = \sup_{x \in \mathcal{X}} W(\daleth_1(x), \daleth_2(x))$.*

*Proof.* Our approach is inspired by the coupling approach proposed by Amortila et al. (2020). Denote by $\Pi(p, q)$ the set of couplings between distributions $p, q$.

Let $\Gamma_{1,x'} \in \Pi(\daleth_1(x'), \daleth_2(x'))$ denote an $\epsilon$-optimal coupling with respect to the Wasserstein distance $W$, in the sense that

$$\int_{\mathscr{P}(\mathcal{X})} \int_{\mathscr{P}(\mathcal{X})} w_d(p, q) \Gamma_{1,x'}(\mathrm{d}p \times \mathrm{d}q) \leq W(\daleth_1(x'), \daleth_2(x')) + \epsilon$$

for arbitrary $\epsilon > 0$. Firstly, we note that $\Gamma_1 \in \Pi((\mathcal{T}^\pi \daleth_1)(x), (\mathcal{T}^\pi \daleth_2)(x))$, where

$$\Gamma_1 = \int_{\mathcal{X}} p^\pi(\mathrm{d}x' \mid x) \left[ (\mathrm{b}_{x,\gamma}, \mathrm{b}_{x,\gamma})_\sharp \Gamma_{1,x'} \right].$$

Here, $(\mathrm{b}_{x,\gamma}, \mathrm{b}_{x,\gamma})_\sharp \Gamma_{1,x'}(A \times B) = \Gamma_{1,x'}(\mathrm{b}_{x,\gamma}^{-1}(A) \times \mathrm{b}_{x,\gamma}^{-1}(B))$ for measurable $A, B \subset \mathscr{P}(\mathcal{X})$. To see this, we note that for any measurable $P \subset \mathscr{P}(\mathcal{X})$,

$$\begin{aligned}
\Gamma_1(P \times \mathscr{P}(\mathcal{X})) &= \int_{\mathcal{X}} p^\pi(\mathrm{d}x' \mid x) \Gamma_{1,x'}(\mathrm{b}_{x,\gamma}^{-1}(P) \times \mathscr{P}(\mathcal{X})) \\
&= \int_{\mathcal{X}} p^\pi(\mathrm{d}x' \mid x) \daleth_1(x')(\mathrm{b}_{x,\gamma}^{-1}(P)) \\
&= \int_{\mathcal{X}} p^\pi(\mathrm{d}x' \mid x) \left[ (\mathrm{b}_{x,\gamma})_\sharp \daleth_1(x') \right](P) \\
&\equiv \left[ (\mathcal{T}^\pi \daleth_1)(x) \right](P),
\end{aligned}$$

so that the first marginal of $\Gamma_1$ is $(\mathcal{T}^\pi \daleth_1)(x)$. Likewise, the second marginal of $\Gamma_1$ is $(\mathcal{T}^\pi \daleth_2)(x)$, confirming that $\Gamma_1$ is a coupling between $(\mathcal{T}^\pi \daleth_1)(x)$ and $(\mathcal{T}^\pi \daleth_2)(x)$. It follows that

$$\overline{W}(\mathcal{T}^\pi \daleth_1, \mathcal{T}^\pi \daleth_2) = \sup_{x \in \mathcal{X}} W((\mathcal{T}^\pi \daleth_1)(x), (\mathcal{T}^\pi \daleth_2)(x))$$

$$\leq \sup_{x \in \mathcal{X}} \int_{\mathscr{P}(\mathcal{X})} \int_{\mathscr{P}(\mathcal{X})} w_d(p,q) \Gamma_1(\mathrm{d}p \times \mathrm{d}q)$$

$$= \sup_{x \in \mathcal{X}} \int_{\mathscr{P}(\mathcal{X})} \int_{\mathscr{P}(\mathcal{X})} \int_{\mathcal{X}} w_d(p,q) p^\pi(\mathrm{d}x' \mid x) \left[(\mathrm{b}_{x,\gamma}, \mathrm{b}_{x,\gamma})_\sharp \Gamma_{1,x'}\right](\mathrm{d}p \times \mathrm{d}q)$$

$$= \sup_{x,x' \in \mathcal{X}} \int_{\mathscr{P}(\mathcal{X})} \int_{\mathscr{P}(\mathcal{X})} w_d(\mathrm{b}_{x,\gamma}(p), \mathrm{b}_{x,\gamma}(q)) \Gamma_{1,x'}(\mathrm{d}p \times \mathrm{d}q)$$

We now claim that $w_d(\mathrm{b}_{x,\gamma}(p), \mathrm{b}_{x,\gamma}(q)) \leq \gamma w_d(p,q)$ for any $p, q \in \mathscr{P}(\mathcal{X})$. To do so, let $\Gamma_2 \in \Pi(p,q)$ be an optimal coupling with respect to $w_d$, which is guaranteed to exist since $(\mathcal{X}, d)$ is a Polish space (Villani, 2008). Define $\Gamma_3 \in \mathscr{P}(\mathcal{X} \times \mathcal{X})$ such that

$$\Gamma_3 = (1 - \gamma)\delta_{(x,x)} + \gamma \Gamma_2$$

It follows that, for any measurable $X \subset \mathcal{X}$,

$$\begin{aligned}
\Gamma_3(X \times \mathcal{X}) &= (1 - \gamma)\delta_{(x,x)}(X \times \mathcal{X}) + \gamma \Gamma_2(X \times \mathcal{X}) \\
&= (1 - \gamma)\delta_x(X) + \gamma \Gamma_2(X \times \mathcal{X}) \\
&= (1 - \gamma)\delta_x(X) + \gamma p(X) \\
&= \mathrm{b}_{x,\gamma}(p)(X)
\end{aligned}$$

which confirms that $\mathrm{b}_{x,\gamma}(p)$ is the first marginal of $\Gamma_3$. The similar argument for the second marginal shows that $\Gamma_3$ is a coupling between $\mathrm{b}_{x,\gamma}(p), \mathrm{b}_{x,\gamma}(q)$. So, we see that

$$\begin{aligned}
w_d(\mathrm{b}_{x,\gamma}(p), \mathrm{b}_{x,\gamma}(q)) &= \inf_{\Gamma \in \Pi(\mathrm{b}_{x,\gamma}(p), \mathrm{b}_{x,\gamma}(q))} \int_{\mathcal{X}} \int_{\mathcal{X}} d(y, y') \Gamma(\mathrm{d}y \times \mathrm{d}y') \\
&\leq \int_{\mathcal{X}} \int_{\mathcal{X}} d(y, y') \Gamma_3(\mathrm{d}y \times \mathrm{d}y') \\
&= (1 - \gamma)d(x, x) + \gamma \int_{\mathcal{X}} \int_{\mathcal{X}} d(y, y') \Gamma_2(\mathrm{d}y \times \mathrm{d}y') \\
&= \gamma w_d(p, q)
\end{aligned}$$

Now, continuing the bound from earlier, we have

$$\begin{aligned}
\overline{W}(\mathcal{T}^\pi \daleth_1, \mathcal{T}^\pi \daleth_2) &\leq \sup_{x,x' \in \mathcal{X}} \int_{\mathscr{P}(\mathcal{X})} \int_{\mathscr{P}(\mathcal{X})} w_d(\mathrm{b}_{x,\gamma}(p), \mathrm{b}_{x,\gamma}(q)) \Gamma_{1,x'}(\mathrm{d}p \times \mathrm{d}q) \\
&\leq \gamma \sup_{x \in \mathcal{X}} \int_{\mathscr{P}(\mathcal{X})} \int_{\mathscr{P}(\mathcal{X})} w_d(p, q) \Gamma_{1,x}(\mathrm{d}p \times \mathrm{d}q) \\
&\leq \gamma \sup_{x \in \mathcal{X}} [W(\daleth_1(x), \daleth_2(x)) + \epsilon] \\
&= \gamma \overline{W}(\daleth_1, \daleth_2) + \gamma \epsilon
\end{aligned}$$

Thus, since $\epsilon > 0$ was arbitrary, the claim follows. $\qquad \square$

**Corollary 1.** *Under the conditions of Proposition 4, if the metric space $(\mathcal{X}, d)$ is compact, then the iterates $(\daleth_k)_{k=1}^\infty$ given by*

$$\daleth_{k+1} = \mathcal{T}^\pi \daleth_k$$

*converge in $\overline{W}$ to $\daleth^\pi$, for any given $\daleth_0 \in \mathscr{P}(\mathscr{P}(\mathcal{X}))^{\mathcal{X}}$.*

*Proof.* Prior to applying the Banach fixed point theorem it is necessary to ensure that $\overline{W}$ is finite on $\mathscr{P}(\mathscr{P}(\mathcal{X}))^{\mathcal{X}}$ to ensure that a fixed point will be reached. Since $\mathcal{X}$ is compact and metrics are continuous, it follows that the metric $d$ is bounded over $\mathcal{X}$, that is,

$$\sup_{x,y \in \mathcal{X}} d(x,y) \leq C < \infty$$

for some constant $C$. As such, the Wasserstein distance $w_d$, as an expectation over distances measured by $d$, is also bounded by $C$, and following the same logic, the metrics $W, \overline{W}$ are bounded by $C$. Then, since $\daleth^\pi = \mathcal{T}^\pi \daleth^\pi$, we have that

$$
\begin{aligned}
\overline{W}(\daleth_k, \daleth^\pi) &= \overline{W}(\mathcal{T}^\pi \daleth_{k-1}, \daleth^\pi) \\
&= \overline{W}(\mathcal{T}^\pi \daleth_{k-1}, \mathcal{T}^\pi \daleth^\pi) \\
&\leq \gamma \overline{W}(\daleth_{k-1}, \daleth^\pi)
\end{aligned}
$$

where the final step leverages the contraction provided by Proposition 4. Then, repeating $k-1$ times, we have

$$
\begin{aligned}
\overline{W}(\daleth_k, \daleth^\pi) &\leq \gamma^k \overline{W}(\daleth_0, \daleth^\pi) \\
&\leq \gamma^k C
\end{aligned}
$$

Since $|\gamma| < 1$ and $C$ is finite, it follows that $\overline{W}(\daleth_k, \daleth^\pi) \to 0$, and since $\overline{W}$ is a metric, $\daleth_k \to \daleth^\pi$ in $\overline{W}$. $\qquad\square$

