# OpenReview forum: "A Distributional Analogue to the Successor Representation"
_ICLR.cc/2024/Conference — Submitted to ICLR 2024_

### Official Review · Reviewer_jmar · 2023-10-29

**Soundness:** 3 good
**Presentation:** 1 poor
**Contribution:** 3 good
**Rating:** 6
**Confidence:** 3

**Summary:**

The paper proposes a distribution version of success feature, coined as the successor measure. The successor measure is defined as, conditioned on an initial state $x$, the Dirac distribution conditioned on a random trajectory induced by the policy. The paper proposes the Bellman backup of such successor measure, and proposes a recipe to estimate such successor measure using the proposed $\delta$-model. Finally the paper evaluates the proposed method on some simple environments.

**Strengths:**

1. The paper proposes a new method for distributional RL by estimating the occupancy measure of the policy, instead of its expectation, which seems like a natural way for the problem (vs. measuring the distribution of the value function.)

2. The paper includes good explanations for the new mathematical definitions which help the reviewer to understand the new concepts.

**Weaknesses:**

1. A few concepts are quite confusing along the paper. First, why is it necessary to redefine the occupancy measure/state(-action) distribution as successor representation or successor measure?

Second, why is it necessary to define the new "random occupancy measure", which is a distribution over distribution, but the inner distribution is just the Dirac distribution which is determined by the realization of the sample of the outer distribution? From my understanding, if one wants a distribution over the value function, suppose they have already have the occupancy measure, they could easily define the distribution of the value function (which is in $\Delta(\mathbb{R}^1)$) by projecting the occupancy measure to the reward vector. I think this is also indicated by prop. 1: to define the distributional return, one needs to take one expectation over the random occupancy measure.

Overall, the theory results are rather limited. Most results seem like straight forward extension from the occupancy measure results to the random occupancy measure version.

2. The significance of section 5.2 is unclear to me. How to tune MMD does not directly relate to the significance of the paper, and many description seems not significant (for example, the detailed description of the median trick).

3. Since the theory contribution is rather limited, the experiment of the paper should be greatly improved. First, the paper should compare with other distribution RL methods, and the current benchmarks are also pretty easy.

**Questions:**

See above.

---

> ### Author Response · Authors · 2023-11-15
>
> We thank the reviewer for their constructive feedback. We suspect that there has been a fundamental misunderstanding about what the DSM represents, and we hope to clarify this here. Particularly, we would like to address the following statements from the review:
>
> > The successor measure is defined as, conditioned on an initial state, the Dirac distribution conditioned on a random trajectory…
>
> > ...why is it necessary to define the new "random occupancy measure", which is a distribution over distribution, but the inner distribution is just the Dirac distribution which is determined by the realization of the sample of the outer distribution?
>
> The DSM is not a Dirac distribution conditioned on a random trajectory, and the “inner distributions” are also not Dirac distributions. The DSM is a probability distribution over distributions over the state space. Notably, the DSM has support on many distributions over the state space, and the distributions on its support are supported generally on the whole state space. Perhaps the reviewer is referring to the $\delta$-model parameterization of the DSM; however, the claims are still not accurate:
> - $\delta$-models are supported on a collection of Diracs, not just one. That is, the $\delta$-models can be supported on arbitrarily many occupancy measures (controlled by the number of model atoms). It is important to note that while these are Diracs, they are Diracs each located on a probability distribution over the state space. So, the locations of the model atom Diracs are infinite dimensional.
> - Each model atom is represented as a generative model of state. Our architecture permits iid sampling from these model atoms, which enables us to leverage the unbiased sample-based MMD estimator to learn the locations of the model atoms.
> - Thus, there are two layers of samples. Sampling from a $\delta$-model produces a generative model (one of the particles of the $\delta$-model), and the generative model samples themselves can draw arbitrarily-many iid state samples to represent the corresponding occupancy measure.
>
> > ...why is it necessary to redefine the occupancy measure/state(-action) distribution…
>
> As the reviewer points out, these concepts are tightly related to the occupancy measure. However, this is standard nomenclature in the literature; the successor representation/measure is often referred to as the discounted occupancy measure.
>
> > From my understanding, if one wants a distribution over the value function, suppose they have already have the occupancy measure, they could easily define the distribution of the value function (which is in P(R)) by projecting the occupancy measure to the reward vector.
>
> > ... First, the paper should compare with other distribution RL methods, …
>
> Our goal is far more ambitious than learning a distribution over the return. Indeed, this can be accomplished with standard distributional RL techniques like you suggest. Moreover, with just the occupancy measure, “projecting onto the reward” only produces a point estimate of the return, not a distribution – indeed, this is how transfer with the SR works, and this process is shown in equation (4) of the paper. Our approach is modeling another “axis of distribution” beyond the SR/occupancy measure, and that is how we can “project” our DSM onto a reward function to produce a distribution over returns. To achieve this functionality, it is necessary to properly model the interaction between random occupancy measures, and no other method has done this.
>
> > The significance of section 5.2 is unclear…
>
> As for the significance of Section 5.2, this is a good question. What distinguishes our problem setting from those where MMD optimization is commonly applied is that, due to bootstrapping, the target distributions that we model are non-stationary, and they can change over time (especially towards the beginning of training). Even with static target distributions, kernel optimization tricks are commonplace as you pointed out. However, existing RL algorithms based on MMD optimization (e.g., Nguyen-Tang et al. (2021), Zhang et al. (2021)) did not address this problem. Especially given the complex space of distributions being modeled in our work, we hypothesized that careful treatment of adaptive kernel parameters could substantially improve the quality of the resulting $\delta$-models. We tested this hypothesis by ablating the adaptive kernel, and found that this was a crucial factor in the success of $\delta$-model training – we invite the reviewer to consult Appendix D (particularly, Figure 9, right), for these results.
>
> ### References
> - Thanh Nguyen-Tang, Sunil Gupta, Svetha Venkatesh. "Distributional reinforcement learning with maximum mean discrepancy." AAAI Conference on Artificial Intelligence (2021).
> - Pushi Zhang, Xiaoyu Chen, Li Zhao, Wei Xiong, Tao Qin, Tie-Yan Liu. "Distributional reinforcement learning for multi-dimensional reward functions." Neural Information Processing Systems (NeurIPS), 2021.

---

> ### Author Response · Authors · 2023-11-18
>
> Thank you for your thorough review and the time you have dedicated to our paper. We have carefully addressed each of your questions and concerns in our rebuttal. We hope our clarifications align with your expectations, and we are eager to answer any further questions you have.
>
> Kind regards, Authors.

---

> > ### Comment · Reviewer_jmar · 2023-11-22
> > **Reviewer response**
> >
> > I appreciate the author's detailed response to address my previous confusion. My major concerns are addressed but I still believe the work would benefit more from more challenging benchmarks and more aims towards better explaining the concepts (I agree the current version is already doing a great job, but sometimes it could still be confusing to the reader). I will raise my score.

---

### Official Review · Reviewer_JnMG · 2023-10-31

**Soundness:** 3 good
**Presentation:** 2 fair
**Contribution:** 2 fair
**Rating:** 6
**Confidence:** 3

**Summary:**

This paper extends the successor representation to distributional RL, and proposes the $\delta$-model that learns the distributional successor measure.

**Strengths:**

The idea of combining distribution RL with successor representation is interesting, which combines the merits of both SR and distributional RL. I think this is a promising and important direction. The theoretical analysis is sound.

**Weaknesses:**

The experiment benchmark (windy grid world) is somehow toy compared with other distributional RL papers.

**Questions:**

1. As for Eq.7, are there any requirements for the reward function $r$ besides deterministic? For instance, does it require $r$ to be linear? BTW, is $r$ assumed to be known in experiments?

2. Can you further discuss the relationship between previous work like Zhang et al 2021b, which also learn multi-dimensional return distribution via MMD?

3. For distributional RL papers, a common benchmark is visual input environments like Atari. I think current benchmark (windy gridworld) is somehow toy. It will be helpful to see experiments with larger scale. What’s more, besides zero-shot policy evaluation, another advantage of SR is multitask training. Can the proposed method be combined with the multitask training?

---

> ### Author Response · Authors · 2023-11-15
>
> We thank the reviewer for their time and positive feedback. We agree with the reviewer that the DSM is a promising and important direction of research. We address the questions outlined by the reviewer below and would like to reiterate that traditional distributional RL and the DSM are solving different problems and aren’t directly comparable. We look forward to engaging with the reviewer during the rebuttal period and hope the reviewer may consider revising their score based on our discourse.
>
>
> > For distributional RL papers, a common benchmark is visual input environments …
>
> Regarding learning a DSM from pixels, one of the reasons we modeled lower-dimensional state spaces is that we believe these results are indicative of how the method will perform if one were to scale to high-dimensional observations for the following reason: We believe learning high-dimensional data distributions isn’t the best way to scale a method like DSM. Compounding the difficulty of learning a distribution over images, DSM must model a distribution over distributions of images. As evidence of this claim, performing dimensionality reduction and operating in a structured low-dimensional latent space seems to yield state-of-the-art results in model-based RL (e.g., Schrittwieser et al. 2020, Hafner et al. 2021) and generative modeling of images (e.g., Esser et al. 2021, Rombach et al. 2022). Given such latent embeddings, our current $\delta$-models can learn the DSM as shown in our paper. Thus, we believe that supplementing DSM with a dimensionality-reduction component is out of the scope of this paper, but would make for a very exciting follow-up work.
>
> > As for Eq.7, are there any requirements for the reward function …
>
> Regarding the requirements of the reward function, this is a strength of the DSM model: all that we require is that the reward function is measurable, which encompasses essentially any reward function. In particular, linearity is not required. The reward function can be a black box; all we need is the ability to query the reward.
>
> > Can you further discuss the relationship between previous work like Zhang et al 2021b …
>
> The work on distributional RL with multi-dimensional reward functions (e.g., Zhang et. al 2021b) is indeed closely related, as the reviewer suggests. Multi-dimensional distributional RL methods can simply model the distribution over random finite-dimensional vectors corresponding to returns from reward signals seen at train time. As such, one could use a multi-dimensional return distribution function like that learned by Zhang et al. to infer the return distributions for linear combinations of the reward functions that are used for training. On the other hand, the DSM is modeling a complex distribution over the simplex on the state space, which requires a novel approach to distribution estimation using generative models, as highlighted by Reviewer 9gwA. Notably, it is this distinction that allows the DSM to predict return distributions for arbitrary bounded measurable reward functions.
>
> >  Is r assumed to be known in experiments?
>
> In our experiments, we assumed access to reward function queries (as is commonplace with SR models). There are several instances where this can be applied, such as goal-reaching,  target-tracking tasks, or continuous control tasks that require engineered reward functions. However, more generally, we can also learn a reward function from data, for instance, by supervised learning from reward samples, using IRL from expert trajectories, or learning a reward function from preferences as in RLHF.
>
> > What’s more, besides zero-shot policy evaluation, another advantage of SR is multitask training. Can the proposed method be combined with the multitask training?
>
> With regard to multitask training, can you please clarify or give an example of the type of setting you had in mind? Like SR methods, our DSM algorithm is policy-dependent but is otherwise task-agnostic. Our response to Reviewer 9gwA also discusses this, but please let us know if this does not address your question.
>
> ### References
> - Robin Rombach, Andreas Blattmann, Dominik Lorenz, Patrick Esser, Björn Ommer. "High-Resolution Image Synthesis with Latent Diffusion Models." IEEE/CVF Conference on Computer Vision and Pattern Recognition (CVPR), 2022.
> - Patrick Esser, Robin Rombach, and Björn Ommer. "Taming transformers for high-resolution image synthesis." IEEE/CVF Conference on Computer Vision and Pattern Recognition (CVPR), 2021.
> - Danijar Hafner, Timothy P. Lillicrap, Mohammad Norouzi, Jimmy Ba. "Mastering Atari with Discrete World Models." International Conference on Learning Representations (ICLR), 2021.
> - Julian Schrittwieser, Ioannis Antonoglou, Thomas Hubert, Karen Simonyan, Laurent Sifre, Simon Schmitt, Arthur Guez, Edward Lockhart, Demis Hassabis, Thore Graepel, Timothy P. Lillicrap, David Silver. "Mastering Atari, Go, chess and shogi by planning with a learned model." Nature Vol. 588, Pages 604-609, 2020.

---

> ### Author Response · Authors · 2023-11-18
>
> Thank you for your thorough review and the time you have dedicated to our paper. We have carefully addressed each of your questions and concerns in our rebuttal. We hope our clarifications align with your expectations, and we are eager to answer any further questions you have.
>
> Kind regards, Authors.

---

> > ### Comment · Reviewer_JnMG · 2023-11-21
> >
> > Thanks for your response. Previously the major concern is the relatively simple experiment setting, as well as other questions. We thank the authors for clarifications, but the concerns on experiment setting remains. However, I would like to raise the score since I think the idea is quite interesting.

---

### Official Review · Reviewer_9gwA · 2023-10-31

**Soundness:** 4 excellent
**Presentation:** 4 excellent
**Contribution:** 3 good
**Rating:** 6
**Confidence:** 2

**Summary:**

This paper introduces a novel distributional RL algorithm that learns the distributional successor measure from training samples. This method allows a clean separation of transition structure, i.e., state occupancy measure, and reward, enabling zero-shot risk-sensitive policy evaluation.

**Strengths:**

1. The proposed method cast the problem of learning the return distribution into learning the distribution of random occupancy measure, decoupling the transition structure and reward functions and thus enabling zero-shot risk-sensitive policy evaluation. In this sense, the proposed method is novel.

2. This paper presented a practical algorithm for training the $\delta$-models, adopting diverse generative models to approximate the distribution of random occupancy measure. The training procedure itself has merit and can be potentially beneficial to the other distribution estimation tasks.

**Weaknesses:**

1. Similar to the conventional successor feature and successor measure, the decoupling of the transition structure and reward functions still assumes a fixed policy and transition dynamic, limiting the usefulness of the proposed method.

2. The usefulness of the distributional SM is quite limited at this point. I would recommend discussing more about the potential applications of the learned distributional SM other than the zero-shot distributional policy evaluation.

**Questions:**

Will the setting of $\gamma = 0.95$ limit the usefulness of the proposed method in practice when we care about the return of a long episode?

---

> ### Author Response · Authors · 2023-11-15
>
> We thank the reviewer for their kind comments and overall positive review. We agree with the reviewer that the DSM is novel and that insights from our $\delta$-model method are useful beyond estimating the DSM. We hope that our response below helps provide additional motivation for the problem we introduce and inspire exciting future work on the DSM. We look forward to an engaging discussion period.
>
> The reviewer makes a fair claim that the theory of DSM (and other SM/SR tasks) is limited to fixed policies. Despite this, the successor representation has garnered significant interest in many facets of RL, for example, representation learning, exploration, and temporal abstractions which could conceivably benefit from a distributional perspective; examples are given in the second paragraph of Section 7.
>
> Our principal goal with this work was to address the difficulty of representing and learning a DSM, which has not been done previously, and we suspect that future work can devise control methods based on the DSM. There are some significant challenges in this direction, which are not specific to the DSM model itself:
> - Even in distributional RL, there is generally no guarantee that return distributions will converge under greedy distributional updates.
> - In risk-sensitive optimal control, it is generally necessary to employ policies that are either non-stationary or non-Markovian, and it is not immediately clear how to model occupancy measures or perform zero-shot evaluation for these types of policies.
> - The work of Touati et. al (2021) introduces the ‘forward-backward’ representation of the successor measure, which enables an agent to learn the SM for all optimal policies simultaneously. But in the distributional case, this necessitates a definition of ‘optimal policy’, which has its own challenges, as mentioned in the previous point.
>
> Solving any of these problems would be an incredible contribution in their own right, and we leave this to future work. That said, we believe there are exciting applications for DSM as presented in this work. Unlike any other algorithm we are aware of, the DSM allows us to generalize across tasks and risk-sensitive criteria which has many potential applications; one such example is the zero-shot risk-sensitive policy selection procedure presented in Section 6.
>
> > Will the setting of $\gamma=0.95$ limit the usefulness of the proposed method in practice when we care about the return of a long episode?
>
> Regarding the limitation of the setting of the discount factor in our experiments, we are happy to report results for larger discount factors – we are running these experiments and will report back. Note, however, that the DSM already handles larger horizons more gracefully than the comparable $\gamma$-models: in their experiments, they could not achieve reasonable performance by training with $\gamma=0.95$ directly; instead, they found it necessary to introduce a heuristic schedule for $\gamma$. Beyond that, even with the discount schedule, they could only achieve good results with a density model, which required further assumptions on the MDP dynamics. On the other hand, thanks to the n-step bootstrapping approach to generative TD learning that we introduce in this work, we can successfully learn $\gamma$- and $\delta$-models for $\gamma=0.95$ (and we will confirm the same for $\gamma=0.99$) without any heuristics like the discount schedule or additional assumptions about the dynamics.

---

> ### Author Response · Authors · 2023-11-18
>
> Thank you for your thorough review and the time you have dedicated to our paper. We have carefully addressed each of your questions and concerns in our rebuttal. We hope our clarifications align with your expectations, and we are eager to answer any further questions you have.
>
> Kind regards, Authors.

---

> ### Author Response · Authors · 2023-11-21
> **Performance with larger discount factor**
>
> In response to the reviewer’s concern that our method may be limited to relatively small discount factors, we launched a $\delta$-model training run in the Pendulum environment with a longer horizon ($\gamma = 0.99$). We used a n-step bootstrapping estimator for the $\delta$-model targets with $n=15$. We found that, even in the early stages of training, the return distribution predictions from the DSM are quite accurate, and have very similar quality to those reported for $\gamma = 0.95$. You may find the corresponding plot in the supplementary material, under the name `dsm-gamma-099.pdf`.
> We hope this alleviates the reviewers’ concerns about the feasibility of training $\delta$-models for longer horizons.

---

### Official Review · Reviewer_xHdx · 2023-11-03

**Soundness:** 3 good
**Presentation:** 3 good
**Contribution:** 3 good
**Rating:** 5
**Confidence:** 2

**Summary:**

They investigate the distributional counterpart of discounted occupancy measures, which they refer to as the distributional success measure (DSM). Leveraging the forward Bellman equation for DSM, they introduce a novel approach for approximating DSM. Their approach entails modeling DSM as a collection of generative models, referred to as $\delta$-models, and employing a maximum mean discrepancy as a loss function to quantify the dissimilarity between "distributions over distributions" stemming from the Bellman equation.

**Strengths:**

Their research problem is both intriguing and relatively novel within the RL community. Their approach to estimating DSM appears to be innovative, although I do have some fundamental concerns that I will address later.

**Weaknesses:**

I have some reservations regarding the current proposed methods.
* Firstly, it remains unclear why equal weights are utilized in equation (10). It seems plausible that we should consider learning these weights.
* Secondly, there is a lack of guidance on selecting the value of  $m$ in equation (10), or determining how many $m$ values are required.
* Thirdly, there appears to be a dearth of theoretical justification for the effectiveness of this modeling and approximation approach. While equation (10) may seem suitable if it exactly represents the true SDM, practical implementation would not align with this ideal scenario.

**Questions:**

* Questions are written in a previous paragraph.

* Let's say we have X = {0,1} (binary). Then, a set of $P(X)$ is parametrized by just one parameter $\mu \in [0,1]$. Sp, $P(P(X))$ is a set of distributions over $[0,1]$. So, if I understand correctly, learning SDM is equivalent to estimating a distribution over  $[0,1]$. Even in this simple case, does the author's approach have any theoretical guarantee? (finite $m$ and equal weights look restrictive?)


### Suggestion for presentation

* Equation (4) may appear somewhat elementary to researcher within the RL community. The current phrasing, such as "Blier et al. 2021 derived this equation...," seems inaccurate and should be revised. I believe that this equation had already gained widespread recognition prior to the work of Blier et al. (2021), as it is commonly featured in numerous standard RL texts and papers.

* As a related point, I generally believe the author should not emphasize whether the definition pertains to discrete or continuous spaces to such an extent. The transformation from a discrete space (when a base measure is the counting measure) to a continuous space (when the base measure is a Lebsgue measure)  is typically straightforward for individuals with a basic understanding of probability. Therefore, in Section 3.2, the statement "though this result is novel in the case of more general state spaces" may be somewhat misleading. I suggest that this aspect should not be categorized as "novel." Instead, I recommend that the author simply highlight the distinctions between the two contexts (SDM and standard distribution RL with rewards).

* It is somewhat unclear which parameters are precisely optimized throughout the entire algorithm. As I understand it, the author optimizes parameters for all generative models simultaneously in equation (16). It would be beneficial to present the algorithm using an algorithmic environment for clarity.

---

> ### Author Response · Authors · 2023-11-15
> **Response to weaknesses**
>
> We thank the reviewer for their insightful comments and engagement with our work, and are glad to hear that the reviewer finds the DSM novel, and the $\delta$-model an innovative means of learning the DSM.
>
> We view the reviewer's main concerns as relating to the possibility of theoretical analysis of $\delta$-models, and the possibility of investigating alternative approximations to the DSM. We believe that both of these are important directions for future work. We also believe that our introduction of the DSM learning problem, together with the empirically successful approach of training $\delta$-models, are already substantial contributions in their own right. We respond to all comments in detail below, and eagerly await further discussion on these points.
>
> > Firstly, it remains unclear why equal weights are utilized in equation (10)...
>
> Our choice to use equally weighted particles to approximately represent probability distributions is motivated by several considerations:
> - Prior work in distributional reinforcement learning. Both the quantile representation introduced by Dabney et al. (2018), and the equally weighted particle representation introduced by Nguyen-Tang et al. (2021) have this form.
> - Prior work in computational statistics. Using equally weighted particles to approximate distributions is also a common approach in computational statistics, such as Wasserstein/MMD gradient flow approximations (see e.g. Chizat and Bach, 2018, Arbel et al., 2019) and herding (Chen et al., 2010).
> - This parameterization can still effectively assign weight to regions of the space of occupancy measures. For instance, if a given measurable set of the space of occupancy measures should have more probability mass, the trained $\delta$-model can (and does) place more model atoms in that measurable set.
>
> We agree with the reviewer that this is not the only choice of approximate representation that could be made. We view this work, introducing the distributional successor measure, as opening up a new research question within distributional RL: how to approximate and learn the DSM effectively. The delta-models introduced in this paper, together with the experimental results, provide an initial answer to this question, and we expect future work to continue to develop understanding in this area.
>
> > Secondly, there is a lack of guidance on selecting the value of m in equation (10)...
>
> Trained $\delta$-models better approximate the true DSM as $m$ increases. Figure 9 in Appendix C shows that the Wasserstein distance between the DSM and the MC return distribution indeed decreases as we increase the number of model atoms.
>
> > Let's say we have $X = 0,1$… [are there] any theoretical guarantee[s]?
>
> The reviewer’s understanding of the DSM here is correct. It is true that we cannot necessarily model the exact DSM with the EWP parameterization – indeed, it is impossible to guarantee this, since the space of DSMs is far too large to search and represent. Our goal here, like in distributional RL, is to find a good approximation to the DSM among a tractable model class (which is the space of EWP representations here). Further discussion about convergence is given below.
>
> > Thirdly, there appears to be a dearth of theoretical justification …
>
> The reviewer is correct that convergence theory for the algorithm has not been established. Our approach is based closely on approximate dynamic programming methods, as well as an MMD-based loss whose behavior in the case of fixed targets has been rigorously analyzed (Arbel et al., 2019). In particular, in our work, we have established that the distributional SM can be computed by distributional dynamic programming (see Appendix F.1), as is the case of the return distribution function in standard distributional RL. What is missing is the behavior of this algorithm under stochastic approximation of the distributional Bellman operator, as well as the consequences of projecting the targets onto the space of EWP representations.
>
> The quantile TD-learning framework and QR-DQN algorithm of Dabney et al. (2018), which is closely related to our approach, had also not established this theory, which was only very recently resolved by Rowland et al. (2023). This has been a recurring trend in distributional RL, where new important algorithms and their analyses have often come separately. Notably, distributional RL with EWP representation and an MMD objective was employed by Nguyen-Tang et al. (2021) and the multidimensional distributional RL work of Zhang et al. (2021), despite the (ongoing) lack of convergence results in both works. We are definitely in agreement with the reviewer that theoretical analysis of these algorithms is a crucial direction for distributional RL as a whole, and we strongly believe that our contribution presents substantial value to the community in providing a starting point for future theoretical and practical innovations, as earlier works in distributional RL have done in the past.

---

> > ### Author Response · Authors · 2023-11-15
> > **Response to suggestions for presentation**
> >
> > To address some specific concerns about the presentation of the paper:
> >
> > > Equation (4) may appear somewhat elementary to researcher within the RL community…
> >
> > We agree with the reviewer that the discrete case, i.e., $V^\pi = (1-\gamma)^{-1}\Psi^\pi r^\pi$ has been broadly recognized, but the continuous measure-theoretic case hasn’t been as widely discussed. We looked into the origins of this equation in the measure-theoretic case, and it seems that both Blier et al. 2021 (Equation 11) and Janner et al. 2020 (Equation 6) had independently introduced this equation. We have revised the text to reflect this. If the reviewer is aware of any prior work beyond these papers, we would greatly appreciate this and will revise the text accordingly.
> >
> > > As a related point, I generally believe the author should not emphasize whether the definition pertains to discrete or continuous spaces to such an extent …
> >
> > This definition, as well as the following definitions, are given in the general measurable state space setting (it applies to both discrete and Polish state spaces). As you suggest, this definition when invoked with a discrete state space endowed with the counting measure reduces to the familiar definition of the SR and related identities. We agree that this can and should be made more clear, and we have adjusted the paper to address this.
> >
> > With regard to the claim of novelty, we agree that perhaps we did not appropriately emphasize how the proposition is novel. As you claim, generalizing notions on discrete spaces to more general measurable spaces (particularly Polish spaces) is a fairly standard exercise. The novelty is in the formulation of the DSM as the fixed point of a distributional recurrence relation. While distributional Bellman equations for the (scalar) return have been known since at least as far back as Sobel (1982), a formulation of reward-agnostic distributional dynamic programming for the distributional successor measure has not previously been proposed. Upon formulating something like Equation 8, we agree that mathematically its correctness is not overly difficult to prove. As you suggest, it is the distinction between this object and the distribution over multidimensional returns which is new, and we have adjusted the text to reflect this.  We greatly appreciate the reviewer’s attention to detail here.
> >
> > > It is somewhat unclear which parameters are precisely optimized throughout the entire algorithm …
> >
> > Yes, you are correct that we are optimizing the parameters of all the model atoms via Equation 16. Furthermore, Algorithm 1 in Appendix 1 details the complete algorithm which we hope can clear up any confusion.
> >
> > ### References
> > - Michael Arbel, Anna Korba, Adil Salim, Arthur Gretton. "Maximum mean discrepancy gradient flow." Neural Information Processing Systems (NeurIPS), 2019.
> > - Lenaic Chizat, Francis Bach. "On the global convergence of gradient descent for over-parameterized models using optimal transport." Neural Information Processing Systems (NeurIPS), 2018.
> > - Yutian Chen, Max Welling, and Alex Smola. "Super-samples from kernel herding." Proceedings of the Conference on Uncertainty in Artificial Intelligence (UAI) (2010).
> > - Derek Yang, Li Zhao, Zichuan Lin, Tao Qin, Jiang Bian, Tieyan Liu. "Fully parameterized quantile function for distributional reinforcement learning." Neural Information Processing Systems (NeurIPS), 2019.
> > - Will Dabney, Mark Rowland, Marc G. Bellemare, Rémi Munos. "Distributional reinforcement learning with quantile regression." AAAI Conference on Artificial Intelligence, 2018.
> > - Thanh Nguyen-Tang, Sunil Gupta, Svetha Venkatesh. "Distributional reinforcement learning with maximum mean discrepancy." AAAI Conference on Artificial Intelligence, 2021.
> > - Michael Janner, Igor Mordatch, Sergey Levine. "$\gamma$-Models: Generative Temporal Difference Learning for Infinite-Horizon Prediction." Neural Information Processing Systems (NeurIPS), 2020.
> > - Léonard Blier, Corentin Tallec, Yann Ollivier. "Learning successor states and goal-dependent values: A mathematical viewpoint." arXiv preprint arXiv:2101.07123 (2021).
> > - Mark Rowland, Rémi Munos, Mohammad Gheshlaghi Azar, Yunhao Tang, Georg Ostrovski, Anna Harutyunyan, Karl Tuyls, Marc G. Bellemare, Will Dabney. "An analysis of quantile temporal-difference learning." arXiv preprint arXiv:2301.04462 (2023).
> > - Pushi Zhang, Xiaoyu Chen, Li Zhao, Wei Xiong, Tao Qin, Tie-Yan Liu. "Distributional reinforcement learning for multi-dimensional reward functions." Neural Information Processing Systems (NeurIPS), 2021.
> > - Matthew J. Sobel. "The Variance of Discounted Markov Decision Processes." Journal of Applied Probability 19.4 (1982): 794-802.

---

> ### Author Response · Authors · 2023-11-18
>
> Thank you for your thorough review and the time you have dedicated to our paper. We have carefully addressed each of your questions and concerns in our rebuttal. We hope our clarifications align with your expectations, and we are eager to answer any further questions you have.
>
> Kind regards, Authors.

---

> > ### Comment · Reviewer_xHdx · 2023-11-21
> >
> > Thank you for the detailed response! I will reconsider the rating.

---

### Author Response · Authors · 2023-11-15
**General Response to Reviewers**

We thank the reviewers for their insights and their valuable feedback. In addition to the individual responses we have made to the reviewers, below is a summary of our feedback and rebuttal.

The principal contribution of this work is the introduction of a new reinforcement learning problem and framework, which models the distribution over state occupancies that a policy induces in a MDP. We provide a characterization of the DSM, as well as a tractable model for representing this object (namely, the $\delta$-model). Further, we provide a practical algorithm for learning $\delta$-models from data which Reviewer 9gwA highlights as a useful technique for distribution estimation problems even beyond the DSM. The resulting framework enables a new form of generalization: beyond generalizing policy evaluation across tasks for the mean return like SR methods, **the distributional successor measure further provides the ability to generalize policy evaluation across tasks and across all statistics of the return**, which Reviewers xHdx, 9gwA, and JnMG agree is novel and of interest to the community.

The main concerns pointed out by the reviewers were related to the validity/flexibility of the proposed learning algorithm, and the choice of benchmarks. We would like to address these two main concerns below:

Regarding the flexibility of the algorithm, we stress that the $\delta$-model parameterization models a rich family of distributions over Janner et al.’s influential $\gamma$-models, which is particularly flexible as the number of model atoms is increased. Further discussion about the flexibility of $\delta$-models and our training procedure is given in response to Reviewer xHdx.

Finally, our suite of benchmarks is meant to illuminate how the DSM paints a much richer picture of the environment by disentangling trajectories from the successor measure – **a feat which no other method in the literature can accomplish**. We follow up with reviewer JnMG with further discussion on future work for scaling the DSM to high-dimensional state-spaces through the use of latent-space models.

---

### Meta-Review · Area_Chair_Bc5e · 2023-12-04

**Metareview:**

This paper considers the distributional RL problem and proposes to study the distributional successor measure (DSM), which is an interesting and new quantity to study in distributional RL. An algorithm for learning the DSM is proposed, which is completely reward-free. However, the paper also has several limitations. First, although the proposed quality is reward-free, it is policy dependent. That is, it only works for problems with fixed policy such as policy evaluation. Second, both theories to justify estimating methods and experiments to show its usefulness are lacking.  In particular, note that both the transition kernel and the DSM have $S(S-1)A$ parameters to estimate (of course in practice people will not estimate in the tabular fashion), if one cannot  show that DSM is easier to estimate than the transition kernel, or estimating DSM has advantage over estimating transition kernel, then it is not clear why one would focus on a more limited quantity that only work with fixed policy. Therefore, without proper theoretical justification, it is not clear why the proposed quantity is more useful than a naive model-based approach that directly estimate the transition kernel.  Overall, despite the paper has interesting contribution, it should be further improved before publication.

**Justification For Why Not Higher Score:**

The paper has several clear weaknesses as stated above.

**Justification For Why Not Lower Score:**

N/A

---

### Decision · Program_Chairs · 2024-01-16

Reject